# A fluorescent sensor for real-time monitoring of DPP8/9 reveals crucial roles in immunity and cancer

Konstantin Weiss[1], Yelizaveta Agarkova[2], Alexandra Zwosta[2], Sarah Hoevel[1], Ann-Kathrin Himmelreich[2], Magdalena Shumanska[3], Julia Etich[4], Gereon Poschmann[5], Bent Brachvogel[4,6], Ivan Bogeski[3], Dirk Mielenz[2], Jan Riemer[1,7]

**Dipeptidyl peptidases 8 and 9 (DPP8/9) are critical for the quality control of mitochondrial and endoplasmic reticulum protein import, immune regulation, cell adhesion, and cell migration. Dysregulation of DPP8/9 is associated with pathologies including tumorigenesis and inflammation. Commonly, DPP8/9 activity is analysed by in vitro assays using artificial substrates, which allow neither continuously monitoring DPP8/9 activity in individual, living cells nor detecting effects from endogenous interactors and posttranslational modifications. Here, we developed DiPAK (for DPP8/9 activity sensor based on AK2), a ratiometric genetically encoded fluorescent sensor, which enables studying DPP8/9 activity in living cells. Using DiPAK, we determined the dynamic range of DPP8/9 activity in cells overexpressing or lacking DPP9. We identified distinct activity levels among melanoma cell lines and found that LPS-induced primary B-cell activation depends on DPP8/9 as the absence of DPP8/9 activity results in apoptotic but not pyroptotic cell death. Consistently, we observed increasing DPP8/9 activity during B-cell maturation. Overall, DiPAK is a versatile tool for real-time single-cell monitoring of DPP8/9 activity in a broad range of cells and organisms.**

## Introduction

The dipeptidyl peptidases 8 and 9 (DPP8 and DPP9, in the following abbreviated as DPP8/9) are serine proteases of the DPPIV family, which are mainly localized to the cytosol but are also present in the nucleus. For DPP9, two different isoforms, DPP9-S and DPP9-L, were identified that show cytosolic and nuclear localization, respectively (Olsen & Wagtmann, 2002; Ajami et al, 2004; Justa-Schuch et al, 2014; Zhang et al, 2015a; Cui et al, 2022). DPP8/9 take important roles in

processes such as the regulation of energy metabolism and organellar protein import, the initiation of immune responses, cell proliferation, differentiation, adhesion, and migration, and apoptosis (Yu et al, 2006; Matheeussen et al, 2013; Zhang et al, 2015a; Justa-Schuch et al, 2016; Tang et al, 2017; Finger et al, 2020). During these processes, DPP8/9 expression often dynamically changes (Chowdhury et al, 2013; Matheeussen et al, 2013; Fernandez-Garcia et al, 2022; Gislason et al, 2024). Moreover, their activity can be modulated through small ubiquitin-like modifier binding, changes in the redox state of its cysteines, or competition between cleavable and non-cleavable substrates (Park et al, 2008; Pilla et al, 2012; Ross et al, 2018; Zolg et al, 2024). Dysregulated DPP8/9 levels and activities have been linked to tumour development, organ fibrosis, the initiation of pyroptosis, and inflammatory responses (Okondo et al, 2017; Tang et al, 2017; Johnson et al, 2018; Huang et al, 2021; Zhang et al, 2021; Harapas et al, 2022). Thus, targeting DPP8/9 is considered an effective therapeutic approach to, for example, sensitize cells for cancer therapies (Bolgi et al, 2022; Benramdane et al, 2023; Bettecken et al, 2023).

DPP8/9 fulfil their functions either by processing specific substrates or in some cases by interacting with proteins without processing them. For instance, DPP8/9 are involved in inflammasome activation by sequestering NLRP1 or CARD8 and mediate NRF2 stability by binding to KEAP1 (Hollingsworth et al, 2021; Sharif et al, 2021; Chang et al, 2023; Tsamouri et al, 2025). During processing, DPP8/9 cleave N-terminal dipeptides off their substrates if proline or alanine residues are at the penultimate position (Ajami et al, 2004; Zhang et al, 2015b; Cui et al, 2022). However, an acidic amino acid residue N-terminal or a proline residue C-terminal to these residues prevents cleavage (Zhang et al, 2015b). Processed DPP8/9 substrates are sensitive to proteasomal degradation via the N-end rule pathway because of the exposure of destabilizing amino acids at their N terminus (Justa-Schuch et al, 2016; Shimshon et al, 2024; Zolg et al, 2024). This is, for example, important for the role of DPP8/

[1]Redox Metabolism Group, Institute for Biochemistry, University of Cologne, Cologne, Germany    [2]Division of Molecular Immunology, Department of Internal Medicine 3, Friedrich-Alexander-Universität Erlangen-Nürnberg and Universitätsklinikum Erlangen, Erlangen, Germany    [3]Molecular Physiology, Institute of Cardiovascular Physiology, University Medical Centre, Georg-August-University, Göttingen, Germany    [4]Department of Pediatrics and Adolescent Medicine, Experimental Neonatology, Faculty of Medicine and University Hospital Cologne, University of Cologne, Cologne, Germany    [5]Institute for Molecular Medicine, Proteome Research, University Hospital and Medical Faculty, Heinrich-Heine-University Düsseldorf, Düsseldorf, Germany    [6]Center for Molecular Medicine Cologne (CMMC), University of Cologne, Cologne, Germany    [7]Cologne Excellence Cluster on Cellular Stress Responses in Aging-Associated Diseases (CECAD), University of Cologne, Cologne, Germany

Correspondence: jan.riemer@uni-koeln.de

9 in surveilling import of precursor proteins into mitochondria and the endoplasmic reticulum, thereby avoiding their potentially deleterious accumulation in the cytosol if they fail to become efficiently imported (Finger et al, 2020; Shimshon et al, 2024). Among the known DPP8/9 substrates is adenylate kinase 2 (AK2) (Wilson et al, 2013; Finger et al, 2020). Mature AK2 is localized to the intermembrane space of mitochondria (IMS) (Fig 1A). It is synthesized at cytosolic ribosomes, and on its way to mitochondria rapidly and efficiently processed by DPP8/9 (Finger et al, 2020). As a consequence, the protein is degraded by the proteasome, if it does not become imported into the IMS. Loss of DPP8/9 or mutating the DPP8/9 processing site in AK2 strongly stabilizes AK2 making this protein an excellent sentinel for DPP8/9 activity (Finger et al, 2020).

Assessing DPP8/9 activity is crucial to understand their diverse physiological roles. Established methods to measure DPP8/9 activity are based on processing of artificial substrates by purified DPP8/9 or by incubation with cell lysates (Geiss-Friedlander et al, 2009; Ross et al, 2018; Donzelli et al, 2023). With these methods, it remains challenging to assess the impact of posttranslational modifications or so far unidentified endogenous modulators of DPP8/9 activity as they might get lost during lysis. Moreover, these assays do not allow for continuous monitoring of DPP8/9 activity in individual, living cells. Lastly, the employed artificial substrates are also targeted by other proteases of the DPP family and are difficult to adapt to more specifically assess the activity of certain dipeptidyl peptidases (Matheeussen et al, 2012). Apart from artificial substrates, also active-site–directed probes for *in-cell* staining of DPP8/9 are available. These allow detecting DPP8/9 by binding to their active site, but lack the ability to quantify DPP8/9 enzymatic activity (Espadinha et al, 2024).

Here, we developed a genetically encoded fluorescent DPP8/9 activity sensor, which overcomes these limitations. To this end, we fused the first 15 amino acids of AK2 to mEGFP, which triggers its efficient DPP8/9-dependent degradation. We combined this DPP8/9-sensitive mEGFP with a red fluorescent protein fused to the same part of AK2 containing a point mutation that renders it DPP8/9-insensitive. Both proteins are expressed from the same vector whereby the expression of the second fusion protein is driven by an internal ribosomal entry site. This effectively generates a ratiometric sensor system that is independent of cellular expression levels and allows the dynamic monitoring of DPP8/9 activity with single-cell resolution. We established the use of this sensor that we termed DiPAK (for DPP8/9 activity sensor based on AK2) for different readout methods including FACS and microscopy and cellular systems. We employed DiPAK in two settings, in which DPP8/9 activity is of particular physiological relevance (Zhang et al, 2013; Justa-Schuch et al, 2016; Cui et al, 2022; Hess et al, 2024). We were able to correlate DPP8/9 activity and expression in two different melanoma cell lines originating from a primary tumour and a brain metastatic lesion. Moreover, using DiPAK we determined fluctuating DPP8/9 activity with a single-cell resolution during B-cell maturation, a process that we show critically relies on DPP8/9.

## Results

### The N terminus of AK2 serves as a DPP8/9-dependent N-degron

AK2 is an IMS protein that depends on three cysteines for mitochondrial import (C40, C42, and C92) (Finger et al, 2020) (Fig 1A). Directly after translation, the start-methionine of AK2 is removed by methionine aminopeptidase, which is followed by DPP8/9-dependent removal of two further amino acids (A2 and P3) (Wilson et al, 2013; Finger et al, 2020). These two processing steps trigger rapid proteasomal degradation of AK2 (Finger et al, 2020). Mutation of the serine that is positioned directly behind the DPP8/9 cleavage site to a proline residue (S4P-AK2) inhibits DPP8/9 processing and consequently results in a strong AK2 stabilization comparable to the one achieved by inhibition of DPP8/9 using the competitive inhibitor 1G244 (Wu et al, 2009; Finger et al, 2020) (Figs 1B and S1A). Mitochondrial import of AK2 can be prevented by mutating C40, C42, and C92 (AK2-3CS). This cytosolic variant of AK2 is very unstable but can be stabilized like WT AK2 by either introducing the S4P mutation (S4P-AK2-3CS), applying 10 μM 1G244 for 16 h (Fig 1C and D), or expressing this AK2 variant in DPP9 knockout cells (DPP9 KO) (Bolgi et al, 2022) (Fig 1E). Because in HEK293 cells, DPP9 protein levels are much higher than DPP8's, only minimal DPP8/9 activity remains in this cell line ([Finger et al, 2020] and Fig S1A). Furthermore, DPP8 protein levels are not significantly changed in DPP9 KO cells when compared to WT (Fig S1C). Of note, S4P-AK2-3CS is not further stabilized by 1G244 treatment or in the DPP9 KO.

The N-terminal 15 amino acids of AK2 are present in an unstructured region (Fig 1F). We hypothesized that this region acts as a degron and that it is sufficient to fuse these amino acids to a fluorescent protein for destabilization (AK2[1–15]-mEGFP; Fig 1G). We generated two fusion protein variants, one with the WT sequence and one bearing the stabilizing S4P mutation. We expressed these fusion proteins in WT and DPP9 KO cells. The expression of AK2(1–15)-mEGFP in DPP9 KO cells or treatment with 1G244 stabilized the fusion protein. Conversely, the S4P mutation rendered the fusion protein stable and no further stabilization was achieved by adding 1G244 or expression in DPP9 KO cells (Figs 1H and S1D).

In summary, the N-terminal 15 amino acids of AK2 can serve as a strong degron that drives degradation of fluorescent proteins but only upon DPP8/9 processing. This can be abrogated by a single point mutation yielding the stable S4P-AK2(1–15)-mEGFP variant.

### Design of the DPP8/9 activity sensor based on AK2 (DiPAK)

To generate a genetically encoded molecular probe capable of monitoring DPP8/9 activity in cell culture, we combined the open reading frames encoding for AK2(1–15)-mEGFP-Strep and S4P-AK2(1–15)-mKate2-HA into one plasmid (Fig 2A). The expression of the DPP8/9-sensitive AK2(1–15)-mEGFP-Strep moiety ("sensor") is thereby mediated by a CMV promoter, which drives strong expression and works well in many cell lines. The expression of the DPP8/9-insensitive AK2(1–15)-S4P-mKate2-HA ("normalizer") is driven by an internal ribosomal entry site to allow internal expression control. Thereby, both genes are always expressed in a similar ratio.

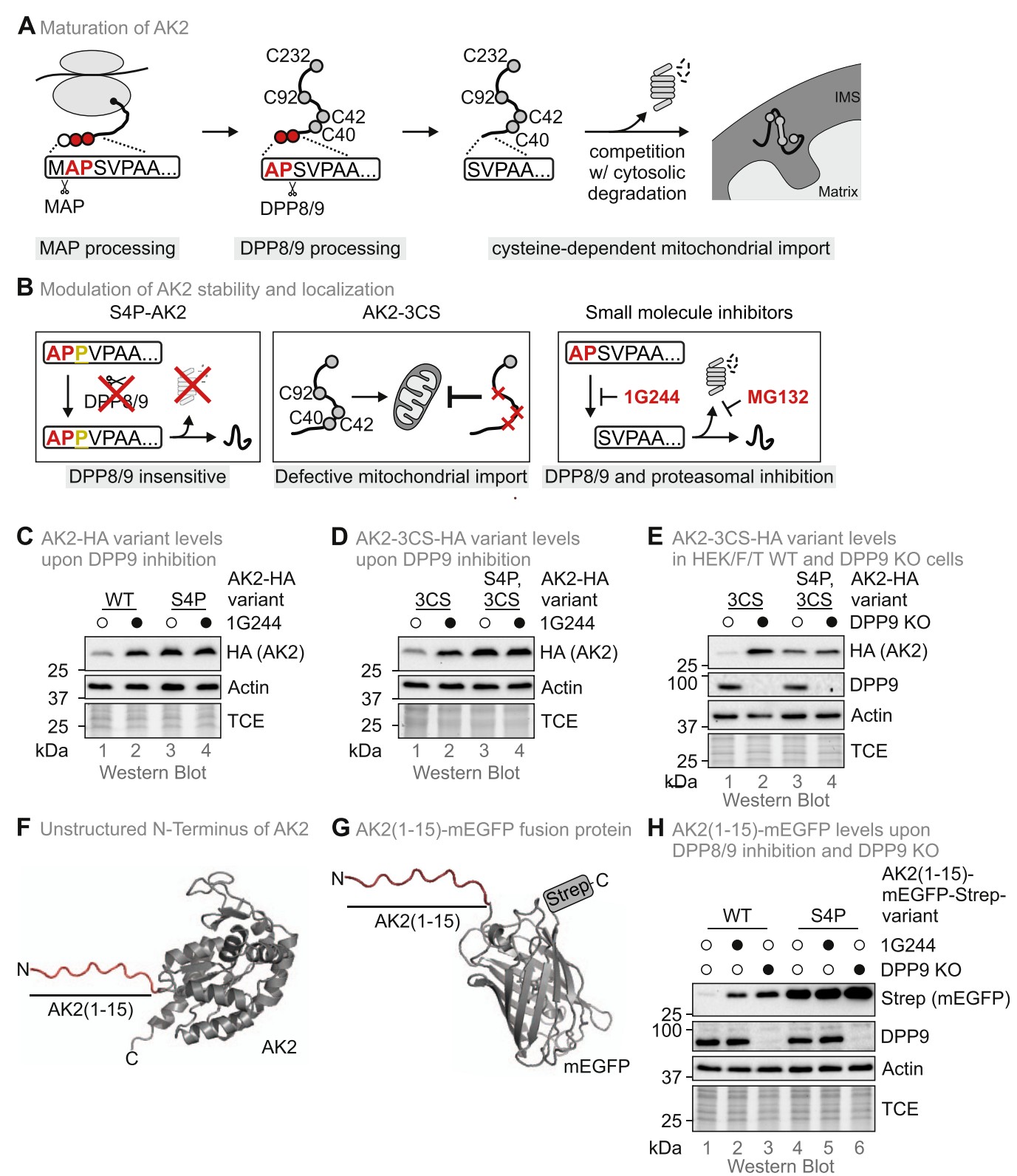

**Figure 1. AK2 N terminus serves as a DPP8/9-dependent degron.**
**(A)** Steps in AK2 maturation. AK2 harbours a DPP8/9 cleavage site at its N terminus. Once the initiator methionine (white circle) is removed by MAP, the following two residues (A2, P3, red circles) can be removed by DPP8/9. This results in a decreased stability of AK2 because of the exposure of degradation signals (N-end rule). AK2 contains four cysteine residues (light grey circles), three of which (C40, C42, and C92) are critical for mitochondrial import of AK2. Import and proteasomal degradation compete with each other determining the amounts of cellular AK2. **(B)** Modulation of AK2 stability and localization. Processing of AK2 by DPP8/9 is abolished by

We confirmed that DiPAK expression did not affect cell growth when compared to the expression of DHFR (Fig S2A). We then showed that upon expression from the DiPAK plasmid, AK2(1–15)-mEGFP-Strep is degraded using an immunoblot chase experiment with the translation inhibitor emetine. S4P-AK2(1–15)-mKate2-HA remained stable under the same conditions (Fig 2B). The ratio between sensor (Strep) and normalizer (HA) signals, which we termed "DiPAK ratio," reflects the information of the sensor system (Fig 2A). During the emetine chase, the DiPAK ratio decreased (Fig 2B).

To further analyse how DiPAK senses changes in DPP8/9 activity, we assessed steady-state DiPAK ratios in HEK293 Flp-In T-REx WT and DPP9 KO cells in combination with treatments with 1G244 or the proteasome inhibitor MG132 that does not directly affect DPP8/9 activity (Fig S1A). In WT cells, almost no AK2(1–15)-mEGFP-Strep signal could be detected at the steady state. Upon DPP8/9 and proteasomal inhibition, the signal strongly increased (Fig 2C). Conversely, in untreated DPP9 KO cells, sensor levels were already elevated and only slightly increased upon DPP8/9 or proteasomal inhibition. S4P-AK2(1–15)-mKate2-HA levels were barely affected by DPP8/9 inhibition and slightly increased upon proteasomal inhibition in both WT and DPP9 KO cells. The DIPAK ratio was about 45 times higher in DPP9 KO cells than in WT cells. Upon DPP8/9 or proteasomal inhibition in WT cells, the DiPAK ratio increased by ca. 40-fold and 25-fold, respectively, when compared to the DMSO control. In contrast, in DPP9 KO cells, the corresponding increases were very minor with only fourfold and twofold, respectively. The DiPAK sensor exhibited a similar behaviour in another cell line, HeLa Flp-In T-REx cells (Fig S2B).

Taken together, DiPAK can serve as a genetically encoded sensor system that allows for a ratiometric monitoring of DPP8/9 activity. The DiPAK ratio thereby inversely correlates with DPP8/9 activity—the higher the ratio, the lower the DPP8/9 activity (Fig 2A).

### DiPAK allows monitoring DPP8/9 activity in intact cells by different methods

Next, we employed DiPAK in HEK293 cells in three different experimental readout settings: fluorescence microscopy, flow cytometry, and a fluorescence plate reader (Fig 3). In fluorescence microscopy, the mEGFP signal was low in WT cells at the steady state

and strongly increased upon treatment of cells with 1G244 and MG132, respectively, whereas mKate2 fluorescence remained apparently constant (Fig 3A). Quantification of the signals confirmed these findings and revealed a significant cell-to-cell heterogeneity of the individual channel intensities (Fig 3B). As expected for a ratiometric sensor, forming the DiPAK ratio lowered the heterogeneity. In WT cells, the DiPAK ratio increased ca. 15-fold and 12-fold upon 1G244 or MG132 treatment, respectively, when compared to the DMSO control (Fig S3A). The DiPAK ratio was also sixfold elevated in DPP9 KO control cells when compared to the WT. The addition of 1G244 or MG132 to DPP9 KO cells resulted in only a slight DiPAK ratio increase (Fig 3B). When comparing the effect of 1G244 to treatments with sitagliptin and KYP-2047, which inhibit DPP4 and PREP, respectively, we did not observe an increase in the DiPAK ratio in WT cells. Similarly, in DPP9 KO cells, the DiPAK ratio was only increased upon treatment with 1G244, indicating that the remaining DPP activity measured is not based on DPP4 or PREP but likely on DPP8 (Fig S3B–D). DPP4 levels in HEK293 cells are relatively low, and thus, we cannot exclude that DPP4 activity in cells containing large amounts of this peptidase could affect DiPAK. PREP is present in HEK293 cells and is not changed in DPP9 KO cells further excluding any effect of this protease on DiPAK.

In flow cytometry, the mEGFP and the mKate2 signal were clearly detected in HEK293 cells. Inhibiting DPP8/9 activity or the proteasome led to a significant increase in mEGFP fluorescence intensity in WT cells, whereas the fluorescence intensity of mEGFP was consistently elevated in DPP9 KO cells. The fluorescence intensity for mKate2 remained stable under all conditions (Fig 3C). Quantification of the DiPAK ratio showed a threefold difference between WT and DPP9 KO cells, and a substantial effect of 1G244 and MG132 treatment on WT cells (up to a 12-fold ratio increase) but not on DPP9 KO cells (Fig S3A).

We also conducted DiPAK measurements on pooled cell populations using a fluorescence plate reader (Fig 3D). In this assay, cells were detached from culture plates and pelleted into 96-well plates for measurement. Again, we found the DiPAK ratio to behave similarly upon DPP8/9 and proteasomal inhibition in WT and DPP9 KO cells (Fig 3D). Collectively, we verified DiPAK sensor system function in different settings with similar dynamic ranges (Fig S3A).

To expand the usability of the DiPAK sensor system even further for varying sets of microscopy set-ups, we created variants with

introducing a single mutation at the AK2 N terminus, S4P (S4P-AK2). A cysteine mutant of AK2, AK2-C40, C42, C92 (AK2-3CS), cannot be imported into mitochondria and, therefore, localizes to the cytosol. Because cytosolic AK2 is target of DPP8/9 processing, it becomes rapidly degraded. Cytosolic AK2 can also be stabilized by treatment of cells with the DPP8/9 inhibitor 1G244 (which also does not affect cell viability at the concentrations employed) (Fig S1B) or the proteasomal inhibitor MG132. **(C)** AK2-HA variant levels in HEK293 Flp-In T-Rex WT cells upon DPP8/9 inhibition. Cells stably expressing either WT AK2-HA or the S4P variant were incubated with 1G244 (10 $\mu$M, 16 h) or DMSO. Subsequently, cell lysates were analysed by reducing SDS–PAGE and immunoblotting. The experiment is representative of three biological replicates. **(D)** AK2-3CS-HA variant levels in HEK293 Flp-In T-Rex WT cells upon DPP8/9 inhibition. Cells stably expressing either AK2-3CS-HA or the corresponding S4P variant were incubated with 1G244 (10 $\mu$M, 16 h) or DMSO. Subsequently, cell lysates were analysed by reducing SDS–PAGE and immunoblotting. The experiment is representative of three biological replicates. **(E)** AK2-3CS-HA variant levels in HEK293 Flp-In T-Rex WT and DPP9 KO cells. AK2-3CS-HA or the corresponding S4P variant was stably and inducibly expressed in WT or DPP9 KO cells. Expression was induced for 24 h using doxycycline. Subsequently, cell lysates were analysed by reducing SDS–PAGE and immunoblotting. The experiment is representative of three biological replicates. **(F)** First 15 amino acids of AK2 are present in an unstructured conformation. The complete AK2 structure was modelled using AlphaFold 2.0 (Jumper et al, 2021). The existing AK2 structure (PDB of human AK2: 2C9Y) does not encompass the unstructured N-terminal part of AK2. **(G)** Design of the AK2(1–15)-mEGFP-Strep fusion protein. The first 15 amino acids of AK2 (AK2[1–15]) serve as an N-degron for mEGFP (PDB of GFP: 1BFP), which is further equipped with a Strep-tag at the C terminus. **(H)** AK2(1–15)-mEGFP-Strep fusion protein levels in HEK293 Flp-In T-REx cells lacking DPP8/9 activity. AK2(1–15)-mEGFP-Strep and the corresponding S4P variant were stably and inducibly expressed in WT or DPP9 KO cells. Expression was induced for 24 h using doxycycline. During the last 16 h, cells were incubated with 1G244 or DMSO. Subsequently, cell lysates were analysed by reducing SDS–PAGE and immunoblotting. The experiment is representative of three biological replicates.
Source data are available for this figure.

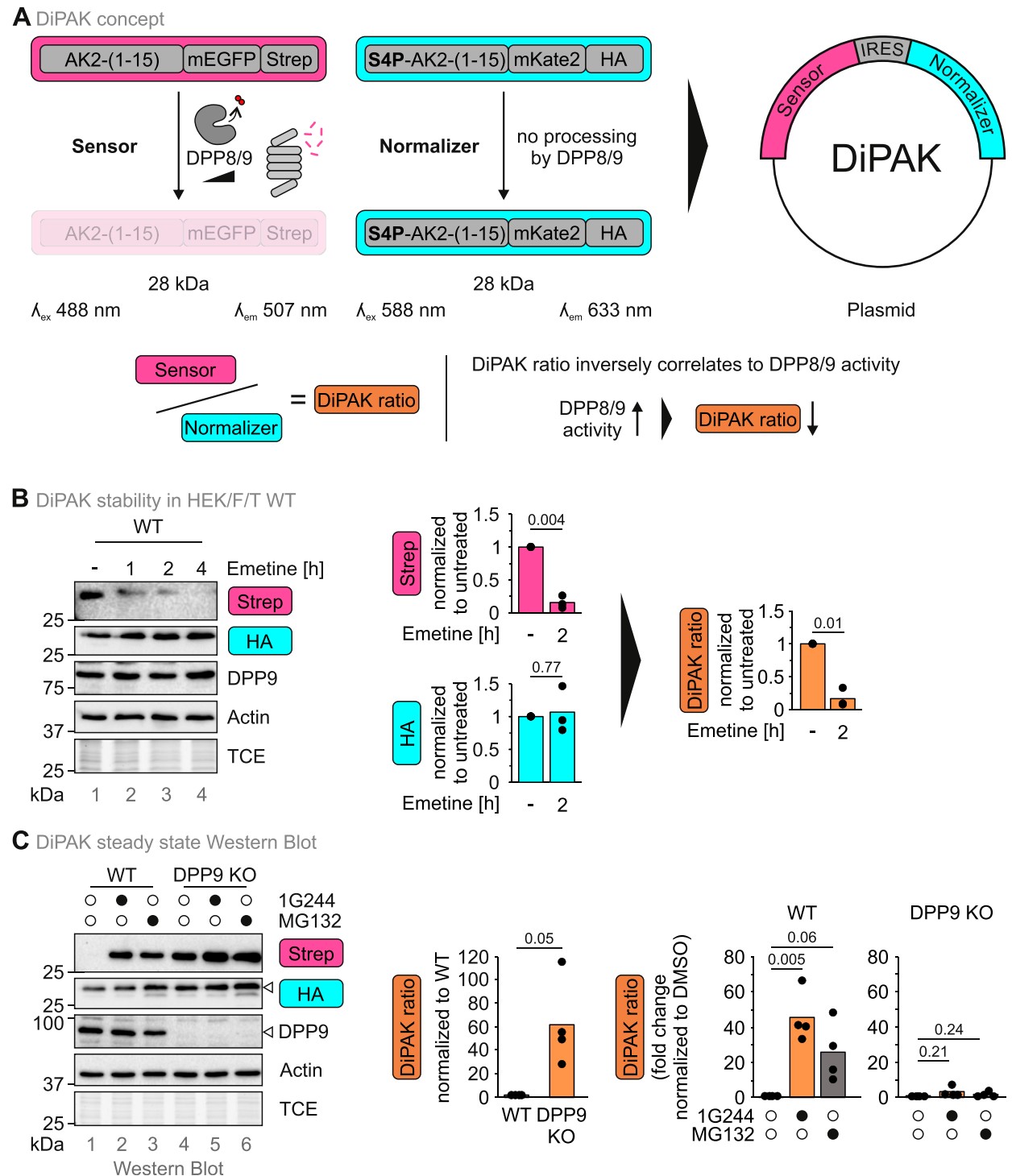

**Figure 2. DiPAK is a genetically encoded ratiometric sensor system for DPP8/9 activity.**
**(A)** Design of the DiPAK sensor system. DiPAK consists of two AK2(1–15)–fluorescent protein fusions. The "sensor unit" harbours the first 15 amino acids of AK2, which can be sequentially processed by MAP and DPP8/9 triggering proteasomal degradation of the sensor. The "normalizer unit" contains the respective S4P variant of AK2, which is insensitive to DPP8/9 processing, thus rendering the normalizer levels independent of DPP8/9 activity. Sensor and normalizer signals can be analysed based on their fluorescence properties (mEGFP or mKate2 fluorescence) or via detection of the protein tags (Strep and HA). The genes encoding both fusion proteins are combined on one plasmid and separated by an internal ribosomal entry site allowing for sensor and normalizer expression in a fixed ratio. The normalized "DiPAK ratio" is calculated by dividing the sensor by the normalizer signals and inversely correlates to DPP8/9 activity. **(B)** Assessment of DiPAK sensor and normalizer stability over time. HEK293 Flp-In T-REx WT cells were treated with the translation inhibitor emetine for up to 4 h. Cell lysates were analysed by reducing SDS–PAGE and immunoblotting. Sensor (Strep) and normalizer (HA) signals were quantified using ImageLab and normalized to actin signals. Protein levels and the mean of three biological replicates were plotted. Statistical comparison was performed using a two-sample *t* test assuming unequal variances. **(C)** Analysis of steady-state DiPAK signals upon DPP8/9 or proteasome

different red fluorescent proteins in the normalizer moiety, which shows similar DiPAK ratio increases upon DPP8/9 inhibition when compared to the respective DMSO control (Fig S4A–C, Table S1). Furthermore, we created a DIPAK version that solely contains mKate2, and no S4P-AK2(1–15) fusion as the normalizer. This DiPAK variant, similar to the others, shows a 12-fold and sevenfold increased DiPAK ratio upon DPP8/9 and proteasomal inhibition, respectively, when compared to the DMSO control in WT cells (Fig S4A and D). Taken together, using DiPAK, we now can quantify DPP8/9-dependent target degradation in living cells in batch or single-cell–based applications.

## DiPAK allows measurements of dynamic DPP8/9 activity changes

Dynamic changes of DPP8/9 activity affect cell differentiation, DNA damage response, mitochondrial protein import, inflammation, and immune responses (Matheeussen et al, 2013; Justa-Schuch et al, 2016; Finger et al, 2020; Hollingsworth et al, 2021; Sharif et al, 2021; Bolgi et al, 2022). To investigate such fluctuations in DPP8/9 activity, we moved away from steady-state assessments and further explored the suitability of the approach to continuously follow DiPAK ratios. Using a fluorescence microscopy reader, we monitored DiPAK ratios for up to 4 h starting directly upon the addition of different 1G244 amounts in WT and DPP9 KO cells (Figs 4A and S5A). To better visualize changes in the DiPAK ratio, we subtracted the DiPAK ratio of the DMSO control from the DiPAK ratio of the 1G244-treated samples. In WT cells, this Δ(DiPAK ratio) increased over time in a 1G244 concentration–dependent manner. Conversely, in DPP9 KO cells, 1G244 did not affect Δ(DiPAK ratio)s. Next, we assessed DiPAK dynamics upon increases in DPP9 activity (Figs 4B and S5B). Different DPP9-S expression levels achieved through expression from a doxycycline-dependent regulatable promoter thereby resulted in decreasing Δ(DiPAK ratio)s. Notably, the changes in the Δ(DiPAK ratio) upon DPP9 overexpression (Fig S5C–E) were smaller compared with changes upon DPP9 loss of function. This might indicate that DPP9 activity is not limiting for substrate processing under normal conditions. In conclusion, DiPAK can be applied to dynamically monitor fluctuating DPP8/9 activity.

## DiPAK reveals different DPP8/9 steady-state activities in distinct melanoma cell lines

DPP9 expression seems to be differentially regulated in various kinds of cancers, and expression levels correlate with patient survival differently ([Tang et al, 2017; Wu et al, 2020; Yokobori, 2020; Hess et al, 2024] and Fig S6A). It currently remains unclear whether the corresponding changes in mRNA levels actually translate to different levels of DPP8/9 activity. We thus employed DiPAK in a

microscopy-based assay to assess DPP8/9 activity in two distinct melanoma cell lines—less aggressive, primary WM1366 and more aggressive, brain metastatic WM3734a cells. We could determine a DiPAK ratio of ca. 3 for WM3734a cells, which is ca. 4.5-fold higher than the mean DiPAK ratio of ca. 0.7 determined for WM1366 cells (Fig 5A). This correlated well with protein levels of DPP9 in those cells (Figs 5B and S6B). Notably, although DPP8 was not detected in the mass spectrometry measurements of both melanoma cell lines, Western blot analysis revealed higher DPP8 levels in WM3734a cells (Fig S6B). DPP8/9 inhibition resulted in a 2.0-fold and 1.6-fold increase of the mean DiPAK ratio in WM1366 and WM3734a cells, respectively, without affecting viability (Figs 5C and S6C).

## DiPAK reveals DPP8/9 activity changes during B-cell maturation

DPP8/9 participate in immunoregulation (Chowdhury et al, 2013; Matheeussen et al, 2013; Waumans et al, 2015; Finger et al, 2020; Suski et al, 2020), and the effects of DPP8/9 on the activation and proliferation of immune cells seem to be mediated by the enzymatic activity of the peptidases (Matheeussen et al, 2013; Waumans et al, 2016; Okondo et al, 2017). For example, in B cells, DPP9 processes splenic tyrosine kinase (Syk), a crucial tyrosine kinase downstream of the B-cell receptor (Justa-Schuch et al, 2016). To explore the role of DPP8/9 activity in B-cell activation, we turned to an in vitro model for mouse B-cell proliferation and differentiation (Fig 6A). We isolated primary murine cells from spleen of mice, sorted B cells, and cultured these primary murine B cells in the presence of different 1G244 amounts upon induction of B-cell proliferation and differentiation by lipopolysaccharide (LPS) (Fig 6B). We found that the number of activated primary B cells was strongly attenuated in the presence of 1G244, and that at concentrations starting from 4 μM 1G244, the numbers of live B cells dropped in the course of LPS treatment because of apoptosis, indicated by Annexin V binding and sub-G1 cell cycle arrest, but not pyroptosis (Figs S7, S8, and S9).

Differentiation of LPS-induced B cells into plasmablasts critically relies on oxidative phosphorylation (Urbanczyk et al, 2022). Thus, we analysed whether DPP8/9 inhibition might be involved in regulation of respiration. We observed dramatic changes in oxygen consumption underlining the important role of DPP8/9 in regulating energy metabolism and mitochondrial biogenesis ([Wilson et al, 2013; Zhang et al, 2015b; Finger et al, 2020], Figs 6C and S7). Basal, ATP-dependent, and maximal oxygen consumption rates were strongly lowered already in the presence of 2 μM 1G244, a concentration that did not completely abolish B-cell proliferation (Fig 6B).

LPS treatment of B cells leads to B-cell activation (TACI+) and, subsequently, to differentiation into plasmablasts (TACI+, CD138+;

---

inhibition. HEK293 Flp-In T-Rex WT cells and corresponding DPP9 KO cells stably expressing DiPAK were treated with DMSO, 10 μM 1G244, or 1 μM MG132. DiPAK expression was induced by the addition of 30 μg/ml cumate for 24 h. During the last 16 h of the induction period, cells were further treated with 1G244 or MG132. DMSO was used as a control treatment. Cells were lysed and analysed by reducing SDS–PAGE and immunoblotting. Sensor (Strep) and normalizer (HA) signals were quantified using ImageLab and normalized to actin signals. The sensor/normalizer ratios and the mean of four biological replicates were plotted. First, the ratios of the DMSO control samples in WT and DPP9 KO cells were compared. Then, the ratio of 1G244 and MG132 samples was normalized to the respective DMSO control for WT and DPP9 KO cells and plotted. Statistical comparison was performed using a two-sample *t* test assuming unequal variances.
Source data are available for this figure.

**A** Changes in DiPAK fluorescence

**B** DiPAK steady state fluorescence microscopy reader

**C** DiPAK steady state flow cytometry

**D** DiPAK steady state fluorescence plate reader

Fig 6A) (Pracht et al, 2017). When discriminating for these markers, we found that among the surviving cells after 3 d of LPS and 1G244 treatment, the share of activated B cells and plasmablasts remained similar (Fig 6D). Only at a 1G244 concentration of 8 µM did the share of activated B cells significantly drop, whereas the established plasmablasts were not affected.

DPP8/9 are up-regulated upon B-cell activation (Chowdhury et al, 2013), and we demonstrated severe effects of 1G244 on B-cell proliferation and function. We thus explored DPP8/9 activity using DiPAK upon in vitro stimulation of primary B cells. On day 0, cells were activated using LPS followed by DiPAK transduction on day 1. On day 2, cells were split into pools that were subjected to 1G244 treatment for 24 h before DiPAK assessment by flow cytometry. In cells that were not treated with 1G244, we found DiPAK ratios to be significantly lowered in plasmablasts compared with activated B cells pointing to higher DPP8/9 activities in plasmablasts (Fig 6E). Cells treated with 1G244 showed—as expected—in general increased DiPAK ratios, but in activated B cells, the DiPAK ratio was much higher compared with plasmablasts to an extent that the DiPAK ratio could essentially be used as a marker to differentiate the cell populations. These data are in line with increased susceptibility of activated B cells to high doses of 1G244 (Fig 6D). Notably, it is unlikely that the 1G244 effects are due to reduced DPP8/9 processing of splenic tyrosine kinase as it is not involved in LPS-elicited B-cell proliferation (Wang et al, 2019). Collectively, with these experiments in primary B cells undergoing proliferation and activation, we demonstrated that DiPAK can be employed in physiological settings and allows differentiation of specific cell populations based on DPP8/9 activity in multiplexing approaches.

# Discussion

DPP8/9 fulfil a variety of physiological functions, many of which are dependent on their catalytic activity. However, little is known about how DPP8/9 activity is regulated in living cells and under which circumstances there might be fluctuations. So far, only in vitro assays have been available to quantitatively assess DPP8/9 activity. They face limitations in investigating different modes of intracellular DPP8/9 activity regulation and in continuously monitoring DPP8/9 activity. Here, we present DiPAK, a genetically encoded fluorescent ratiometric sensor system that addresses these limitations and allows continuous monitoring of DPP8/9 activity in multiplexing experiments in the cellular context.

The DiPAK ratio is a suitable readout for changes in DPP8/9 activity that can be assessed in different measurement settings. However, like for other protease sensors (Grotzke et al, 2013; Kim et al, 2021; Sadoine et al, 2021), its dynamics depend on the constant synthesis and proteasomal degradation of the sensor requiring suitable controls for the functionality of these systems. The fluorescence signal of the DPP8/9-insensitive moiety of the sensor serves as an internal control for translation (Fig S5A and B), whereas treatment with MG132 or emetine can help to assess proteasomal functionality. Interestingly, incubation of cells with the proteasomal inhibitor MG132 did not result in DiPAK ratio changes to the same extent as observed upon 1G244 treatment. We found that upon treatment with MG132, not only the DPP8/9-sensitive sensor signal but also the DPP8/9-insensitive normalizer signal was increased. This resulted in a comparatively lowered ratio and most likely reflects that proteasomal inhibition stabilizes all cytosolic proteins, which also seems to be the case for the normalizer of DiPAK. Interestingly, treatments with the DPP8/9 inhibitor 1G244 did lead to increases in the DiPAK ratio not only in HEK293 WT cells, but also—to a minor extent—in HEK293 DPP9 KO cells (Fig 3). This effect can presumably be attributed to the low amount of DPP8 present in HEK293 cells that we have shown to also impact AK2 levels (Finger et al, 2020), but not to PREP and DPP4, two other proteases that potentially could target AK2. In addition to decreases in DPP8/9 activity, also increases in their activity can be monitored. However, it appears that the DiPAK sensor is particularly susceptible to DPP8/9 inhibition and less so to overexpression (Figs 4 and S5D and E) likely because endogenous DPP8/9 levels are not limiting for cytosolic AK2 processing in HEK293 cells. In conclusion, we provide here a DPP8/9 activity sensor system that through its coupling of DPP8/9-sensitive and DPP8/9-insensitive moieties allows the

**Figure 3. DiPAK ratio can be determined using different readout methods.**
**(A)** Fluorescence microscopy images of DiPAK in HEK293 Flp-In T-Rex WT and DPP9 KO cells upon DPP8/9 or proteasomal inhibition. Cells stably expressing DiPAK and uninduced controls were treated with DMSO, 10 µM 1G244, or 1 µM MG132 for 16 h, respectively. Fluorescence images of mEGFP ($\lambda_{ex}$ = 469 ± 17.5/$\lambda_{em}$ = 525 ± 19.5 nm), mKate2 ($\lambda_{ex}$ = 586 ± 7.5/$\lambda_{em}$ = 647 ± 28.5 nm), and DAPI ($\lambda_{ex}$ = 353/$\lambda_{em}$ = 465 nm) signals were acquired by fluorescence microscopy. Representative images of experiments performed in up to three biological replicates. The bar scales to 100 µm. **(B)** Quantitative single-cell analysis of steady-state DiPAK signals using a fluorescence microscopy reader. HEK293 Flp-In T-Rex WT and DPP9 KO cells stably expressing DiPAK were treated with DMSO, 10 µM 1G244, or 1 µM MG132 for 16 h. Sensor (mEGFP) and normalizer (mKate2) fluorescence signals were assessed by fluorescence microscopy. Single-channel data and the sensor/normalizer ratios are displayed in boxplots—each data point refers to a single cell. The boxplots visualize the median, and the 25[th] (lower hinge) and the 75[th] (upper hinge) percentile. The whiskers extend from the lower and upper hinge to the largest and smallest value, respectively, but no further than 1.5*IQR (interquartile distance) from the respective hinge. Statistical comparison was performed using Welch's *t* test. Total numbers of analysed cells from three biological replicates: WT DMSO: 50; WT 1G244: 75; WT MG132: 64; DPP9 KO DMSO: 66; DPP9 KO 1G244: 66; DPP9 KO MG132: 42. **(C)** Quantitative single-cell analysis of steady-state DiPAK signals by flow cytometry. HEK293 Flp-In T-Rex WT and DPP9 KO cells stably expressing DiPAK were treated with DMSO, 10 µM 1G244, or 1 µM MG132 for 16 h. Sensor (mEGFP) and normalizer (mKate2) fluorescence signals were assessed by flow cytometry. Clusters of DMSO-, 1G244-, and MG132-treated samples are depicted in black, orange, and grey, respectively. Sensor/normalizer ratios are displayed in boxplots. The boxplots visualize the median, and the 25[th] (lower hinge) and the 75th (upper hinge) percentile. The whiskers extend from the lower and upper hinge to the largest and smallest value, respectively, but no further than 1.5*IQR (interquartile distance) from the respective hinge. Statistical comparison was performed using Welch's *t* test. Total numbers of analysed cells from two biological replicates: WT DMSO: 7259; WT 1G244: 7515; WT MG132: 5013; DPP9 KO DMSO: 8035; DPP9 KO 1G244: 8230; DPP9 KO MG132: 5264. **(D)** Quantitative analysis of steady-state DiPAK signals using a fluorescence plate reader. HEK293 Flp-In T-Rex WT and DPP9 KO cells stably expressing DiPAK were treated with DMSO, 10 µM 1G244, and 1 µM MG132 for 16 h. Sensor (mEGFP) and normalizer (mKate2) fluorescence signals were assessed using a fluorescence plate reader. Single-channel data and the sensor/normalizer ratios are displayed in bar charts—each data point refers to one of 14 single measurements from two biological replicates. Statistical comparison was performed using Welch's *t* test.
Source data are available for this figure.

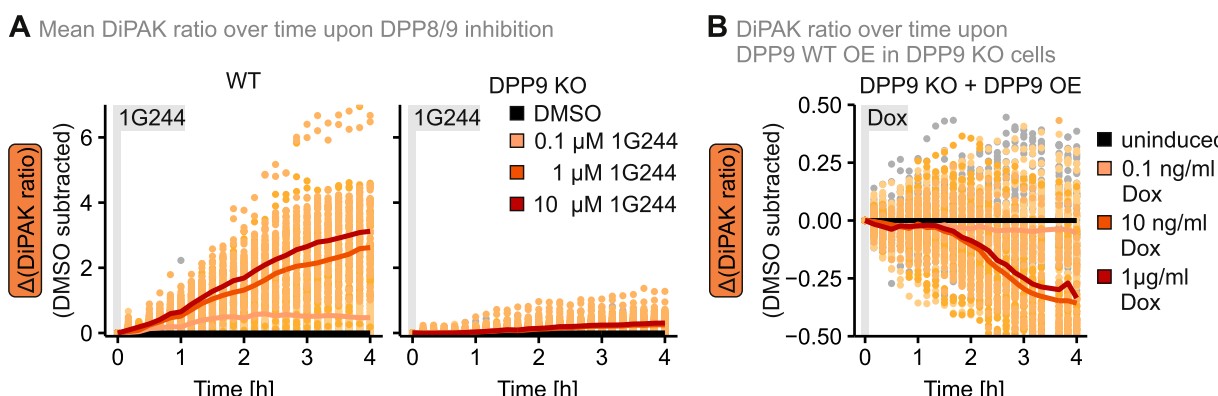

**Figure 4. DiPAK allows continuously monitoring DPP8/9 activity in living cells.**
**(A)** Changes in DiPAK ratio over time upon DPP8/9 inhibition. The expression of DiPAK in HEK293 Flp-In T-Rex WT and DPP9 KO cells stably expressing the sensor system was induced by the addition of 30 μg/ml cumate for a total of 24 h. Then, cells were treated with DMSO and different amounts of 1G244 for up to 4 h and DiPAK fluorescence was continuously monitored using a multiwell microscopy reader. The sensor (mEGFP)/normalizer (mKate2) ratio is depicted in black and different shades of orange for DMSO- and 1G244-treated samples, respectively. Single-cell data are represented as dots, and the mean is represented as a solid line. Total numbers of analysed cells from three biological replicates: WT DMSO: 36; WT 0.1 μM 1G244: 47; WT 1 μM 1G244: 59; WT 10 μM 1G244: 57; DPP9 KO DMSO: 54; DPP9 KO 0.1 μM 1G244: 68; DPP9 KO 1 μM 1G244: 71; DPP9 KO 10 μM 1G244: 94. **(B)** Changes in the DiPAK ratio over time upon DPP9-S WT overexpression. The expression of DiPAK in HEK293 Flp-In T-Rex DPP9 KO + DPP9-S WT cells stably expressing the sensor system was induced by the addition of cumate for a total of 24 h. Then DPP9-S WT expression was induced by doxycycline (Dox) addition for up to 4 h, and DiPAK fluorescence was continuously monitored using a multiwell microscopy reader. The sensor (mEGFP)/normalizer (mKate2) ratio is depicted in black and different shades of orange for DMSO- and doxycycline-treated samples, respectively. Single-cell data are represented as dots, and the mean is represented as a solid line. Total numbers of analysed cells from three biological replicates: DMSO: 51; 0.1 ng/ml Dox: 47; 10 ng/ml Dox: 51; 1 μg/ml Dox: 36. Source data are available for this figure.

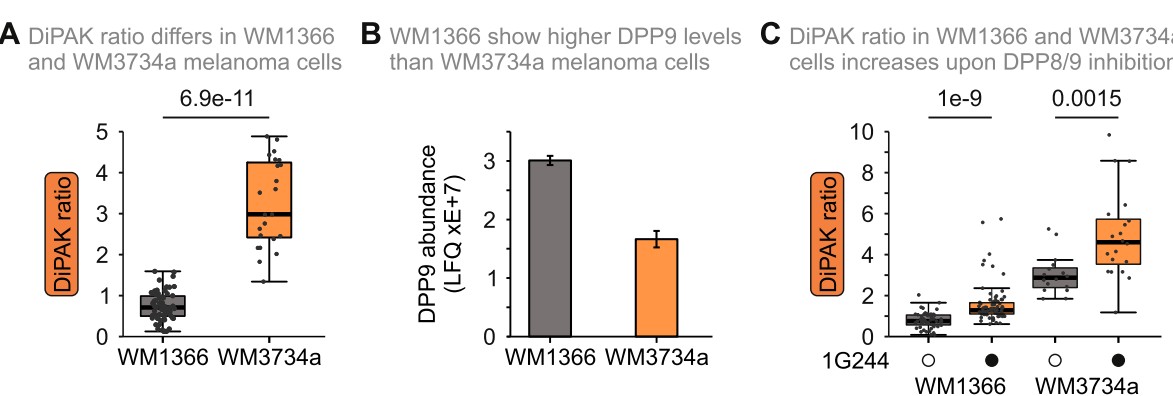

**Figure 5. DiPAK reveals different DPP8/9 activities in distinct melanoma cell lines.**
**(A)** DiPAK measurement in primary and metastatic melanoma cells by fluorescence microscopy. WM1366 and WM3734a cells were transiently transfected with DiPAK (with mScarlet-I in the normalizer unit) for 48 h. Steady-state DiPAK fluorescence signals were assessed by live single-cell fluorescence microscopy. Sensor/normalizer ratios quantified in primary melanoma cells were lower compared with those of metastatic melanoma cells, indicating higher DPP8/9 enzymatic activity. The sensor/normalizer ratios are displayed in a boxplot—each data point refers to the ratio of a single cell. Statistical comparison was performed using Welch's *t* test. Total numbers of analysed cells from two biological replicates: WM1366: 62; WM3734a: 23. **(B)** Abundance of DPP9 in primary and metastatic melanoma by quantitative mass spectrometry. The primary melanoma cell line WM1366 shows higher DPP9 protein levels compared with the metastatic cell line WM3734a. The plot represents the mean ± SD of relative label-free quantification of DPP9 abundance from three technical replicates (Poschmann et al, 2022). **(C)** DiPAK measurement upon DPP8/9 inhibition in primary and metastatic melanoma cells by fluorescence microscopy. WM1366 and WM3734a cells were transiently transfected with DiPAK (with mScarlet-I in the normalizer unit) for 48 h and treated with DMSO or 10 μM 1G244 for the last 16 h of the transfection time. Steady-state DiPAK fluorescence signals were assessed by live single-cell fluorescence microscopy. Sensor/normalizer ratios quantified in primary melanoma cells were lower compared with those of metastatic melanoma cells, indicating higher DPP8/9 enzymatic activity. The sensor/normalizer ratios are displayed in a boxplot—each data point refers to the ratio of a single cell. Statistical comparison was performed using Welch's *t* test. Total numbers of analysed cells from two biological replicates: WM1366 DMSO: 51; WM1366 1G244: 82; WM3734a DMSO: 16; WM3734a 1G244: 20. Source data are available for this figure.

focussed and dynamic analysis of an important cytonuclear processing step.

We applied our carefully established DiPAK sensor in two different physiologically relevant settings (human melanoma and murine B cells), in which it faithfully reflected on DPP8/9 activity. For different cancer cells, DPP8/9 activity might be indicative of

patient survival. For example, previous studies reported associations of DPP9 expression with patient survival in non–small-cell lung cancer, breast cancer, colorectal cancer, and oral squamous cell carcinoma (Tang et al, 2017; Wu et al, 2020; Yokobori, 2020; Hess et al, 2024). Notably, expression levels thereby did differentially correlate with survival. In some cancers, increased DPP8/9

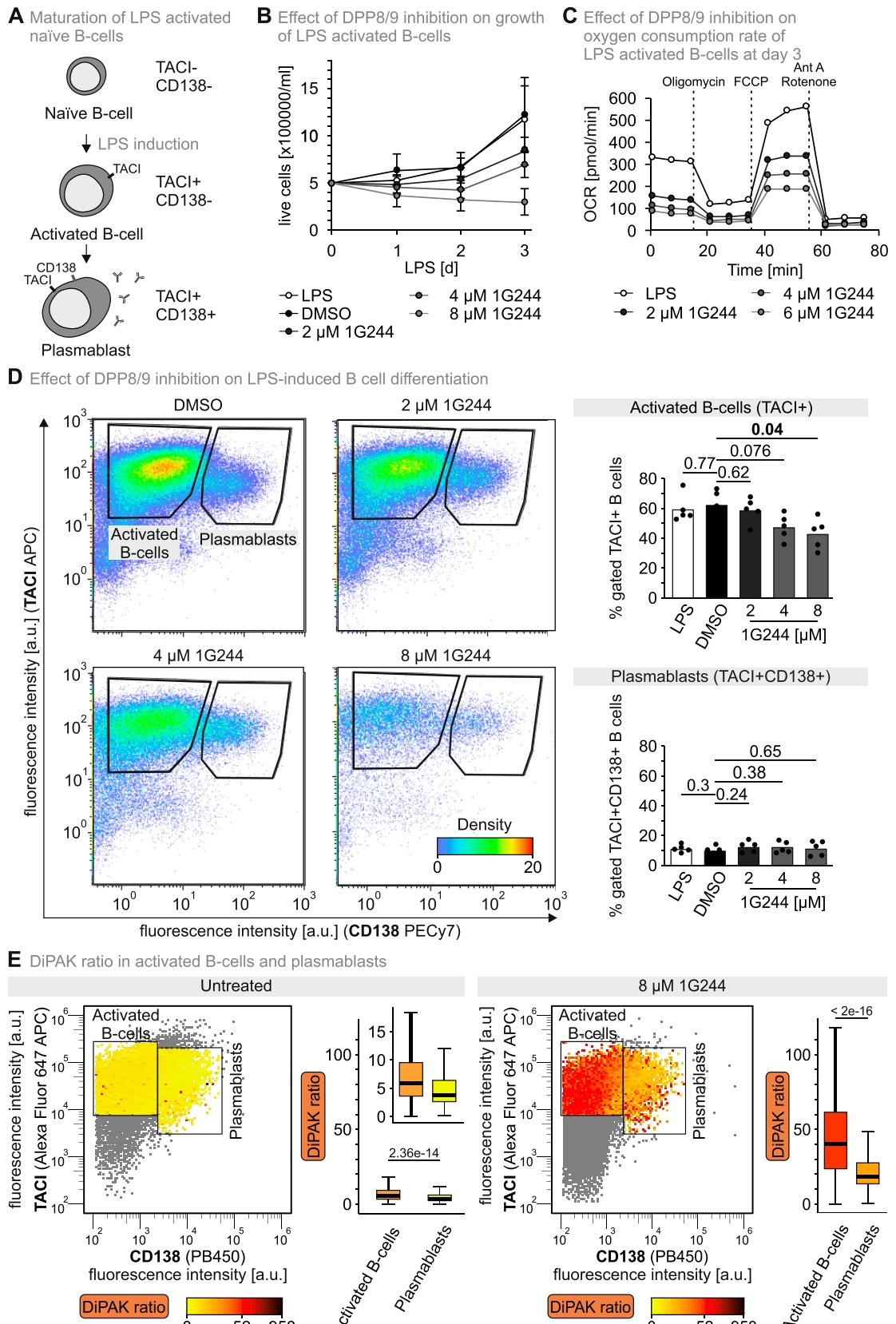

**A** Maturation of LPS activated naïve B-cells

**B** Effect of DPP8/9 inhibition on growth of LPS activated B-cells

**C** Effect of DPP8/9 inhibition on oxygen consumption rate of LPS activated B-cells at day 3

**D** Effect of DPP8/9 inhibition on LPS-induced B cell differentiation

**E** DiPAK ratio in activated B-cells and plasmablasts

transcript levels indicated lowered patient survival, whereas in others, the situation was vice versa. A fluorescent activity sensor like DiPAK might in future be used to evaluate the metastatic, that is, aggressive, potential of cancer cells in vitro but also in 3D cell culture, organoid and spheroid, models (Bettecken et al, 2023; Hess et al, 2024). DiPAK also reflected on a single-cell basis of DPP8/9 activity differences in activated B cells and plasmablasts that correlated well with other markers and provided evidence for the importance of DPP8/9 in those cells. Specifically, activated, highly proliferative TACI+CD138- B cells showed less DPP8/9 activity than plasmablasts, explaining their increased sensitivity towards DPP8/9 inhibition. Given the role of DPP8/9 in regulation of metabolic processes and our finding that respiration of LPS-induced B cells is reduced upon DPP8/9 inhibition, this underlines the concept that oxidative catabolic pathways support LPS-elicited B-cell expansion (Urbanczyk et al, 2022).

It will be exciting to follow up on these initial applications and employ DiPAK to understand, for example, intracellular post-translational modification–driven DPP8/9 activity regulation events or use the sensor design as a basis for further ratiometric protease activity sensors with different specificities and characteristics.

# Materials and Methods

## Plasmids, cell lines, and chemical treatments of cells

For plasmids and cell lines used in this study, see Tables S1 and S2. All cell lines were cultivated using DMEM (Cat# 11965092; Thermo Fisher Scientific) complete containing 4.5 g/l glucose, L-gluta-mine, and 10% FCS (Cat# P40-37500; PAN Biotech), at 37°C under 5% $CO_2$.

For emetine chase experiments, 500,000 HEK293 cells stably expressing DiPAK were seeded onto poly-L-lysine–coated six-well plates. One day after seeding, DiPAK expression was induced by the addition of 30 $\mu$g/ml cumate for 24 h. During the last 4 h of DiPAK expression, cells were treated for indicated times with 100 $\mu$g/ml emetine (dissolved in water). After each time point, cells were

washed with 1 ml ice-cold PBS and harvested in reducing Laemmli buffer. The samples were analysed by SDS–PAGE and subsequent Western blotting.

For DMSO, 1G244, and MG132 treatment, cells were incubated with 1 $\mu$l/ml DMSO, 10 $\mu$M 1G244, and 1 $\mu$M MG132, respectively, for 16 h if not indicated differently. After treatment, cells were analysed by fluorescence microscopy, using a fluorescence plate reader, SDS–PAGE, and subsequent Western blot or flow cytometry.

## Generation of stable inducible HEK293 cell lines

The Flp-InT-REx system (Invitrogen) was used to create stable, inducible cell lines expressing DPP9-S WT, DPP9-L WT, DHFR, or AK2 variants. DPP9-S WT, DPP9-L WT, DHFR, or AK2 variants were cloned into the pcDNA5/FRT/TO vector and cotransfected with the pOG44 vector into HEK293 Flp-In T-REx cells using the transfection reagent FuGENE HD (Cat# E2311; Promega) according to the manufacturer's guideline. For selection of positive clones, DMEM complete containing 100 $\mu$g/ml hygromycin and 10 $\mu$g/ml blasticidin was used. Unless otherwise stated, DPP9 expression was induced by the addition of DMEM complete containing 1 $\mu$g/ml doxycycline for the time indicated.

The PiggyBac system (System Biosciences) was used to create stable, inducible cell lines expressing DiPAK variants. DiPAK variants were cloned into the PiggyBac vector and cotransfected with a vector harbouring the Super PiggyBac transposase into HEK293 cells using the transfection reagent FuGENE HD according to the manufacturer's guideline. For selection of positive clones, DMEM complete containing 2 $\mu$g/ml puromycin was used. Unless otherwise stated, DiPAK expression was induced by the addition of DMEM complete containing 30 $\mu$g/ml cumate for 24 h.

## Transient transfection of HEK293 WT cells

For transient transfection of DiPAK variants, $5 \times 10^3$ HEK293 WT cells were seeded onto poly-L-lysine–coated 96-well plates. 1 d after seeding, cells were transfected with 0.05 $\mu$g of DiPAK containing pcDNA3.1(+) plasmids/well using 0.15 $\mu$l polyethylenimine. Cells were incubated for 24 h.

---

**Figure 6. DiPAK reveals different responses of activated B cells and plasmablasts to DPP8/9 inhibition.**
**(A)** Maturation of naïve B cells upon the addition of LPS. Schematic of B-cell stages classified by surface marker (TACI; CD138) expression in LPS cultures. **(B)** DPP8/9 inhibition reduces growth of LPS-activated B cells. Murine splenic B cells were activated with LPS and cultured in the presence of DMSO or increasing concentrations of 1G244 dissolved in the same amount of DMSO for 3 d. Cells were counted each day and counterstained with trypan blue to distinguish live and dead cells. Data represent the mean –/+ SD from five biological replicates. **(C)** DPP8/9 inhibition reduces the oxygen consumption rate of LPS-activated B cells. Murine splenic B cells activated for three days with LPS at indicated 1G244 concentrations were subjected to a Mito Stress Test in a Seahorse XF96 analyser. At indicated time points, oligomycin, carbonyl cyanide-p-trifluoromethoxyphenylhydrazone (FCCP), or rotenone/antimycin A were added. The graph is a representative example of three independent cultures measured. Statistical analysis of all replicates is shown in Fig S7. **(D)** Cell numbers of plasmablasts are less reduced by increasing strengths of DPP8/9 inhibition than cell numbers of activated B cells. Representative flow cytometric analysis of DMSO- or 1G244-treated murine splenic B cells cultured for three days with LPS. Numbers indicate frequencies (% gated of live cells). The charts represent the frequencies of TACI+CD138- and TACI+CD138+ cells at different concentrations of 1G244. Each dot represents a culture from a distinct mouse. Statistical comparison was performed using Welch's t test. A representative pregating strategy for flow cytometric analysis of activated B cells is displayed in Fig S7. **(E)** DiPAK ratio in plasmablasts is lower than in activated B cells. Activated murine splenic B cells were transduced with the DiPAK sensor. 48 h later, the DiPAK ratio was determined by flow cytometry of either untreated cells or cells previously treated for 12 h with 8 $\mu$M 1G244. The DiPAK ratio is represented colour-coded in the two populations as indicated. Boxplots represent the distribution of the DiPAK ratio in gates of activated B cells and plasmablasts, respectively. The boxplots visualize the median, and the 25th (lower hinge) and the 75th (upper hinge) percentile. The whiskers extend from the lower and upper hinge to the largest and smallest value, respectively, but no further than 1.5*IQR (interquartile distance) from the respective hinge. Statistical comparison was performed using Welch's t test. Numbers of analysed cells: activated B cells untreated: 18,496; plasmablasts untreated: 4,074; activated B cells 1G244: 25,941; plasmablasts 1G244: 3,009. Source data are available for this figure.

## Western blot image acquisition

The immunoblotting images were detected using ChemiDoc Touch Imaging System (Bio-Rad). Image analysis and quantification were performed using ImageLab 5.2 (Bio-Rad).

## GP-AMC activity assay

To measure DPP activity in cell lysates, $2 \times 10^6$ cells of the respective cell line were seeded onto 10-cm dishes. Where applicable, DPP9-S WT expression was induced using 1 µg/ml doxycycline for the time indicated. 2 d after seeding, the cells were washed and harvested in 5 ml ice-cold PBS and centrifuged for 3 min at 700$g$ and 4°C. The cell pellet was resuspended in 1 ml transport buffer (TB; 20 mM Hepes, 110 mM potassium acetate, 2 mM magnesium acetate, 0.5 mM EGTA, 0.02% Tween-20, pH 7.3) per dish and lysed for 30 min on ice. The cell lysates were sonicated and centrifuged for 30 min at 18,000$g$ and 4°C. DMSO, 1G244, ethanol, and MG132 samples were pretreated with 10 µl/ml DMSO, 250 nM or 10 µM 1G244, 10 µl/ml ethanol, 1 µM MG132, 10 µM sitagliptin, and 10 µM KYP-2047, respectively, for 15 min at 25°C before the measurement.

Before each measurement, 25 µl TB and 25 µl lysate (or 22.2 µl TB and 27.8 µl of preincubated sample) were mixed in a 96-well plate. Shortly before measurement, 50 of 500 µM GP-AMC was added to start the reaction. The fluorescence signal was measured using a CLARIOstar (BMG) plate reader with 350- to 35-nm (excitation) and 465- to 35-nm (emission) filters and CLARIOstar software. Data were acquired by measuring every 30s for 30 min at 25°C. The fluorescence signal was normalized to total protein levels, which were determined using BCA Reagent ROTI Quant Assay according to the manufacturer's instructions. Data were analysed using Microsoft Excel 2019.

## Cell proliferation assay

For analysis of cell proliferation, 15,000 HEK293 WT or DPP9 KO cells were seeded onto a poly-L-lysine–coated 48-well plate. After incubation at 37°C for about 8 h, DiPAK or DHFR expression was induced by the addition of 30 µg/ml cumate or 1 µg/ml doxycycline. Right after induction, proliferation analysis was started: every 6 h, each well was imaged using the CytoSMART Omni and cell confluency was determined for 5 d.

## Cell viability assay

For analysis of HEK293, WM1366, and WM3734a cell viability, cells were seeded onto poly-L-lysine–coated 96-well plates and treated with DMSO, 1 µM 1G244, or 10 µM 1G244 for 4–96 h. After the treatment, cell viability was analysed PrestoBlue Cell Viability Reagent (Thermo Fisher Scientific) according to the manufacturer's instructions. In brief, media were replaced by media containing 10% PrestoBlue Cell Viability Reagent. After incubation at 37°C for 1.5 h, fluorescence ($\lambda_{ex} = 560/\lambda_{em} = 590$ nm) was measured. For analysis, fluorescence values of samples without cells were subtracted and resulting fluorescence intensities were compared with respective DMSO-treated samples.

## Fluorescence microscopy

For analysis of DiPAK fluorescence, $5 \times 10^4$ HEK293 WT or DPP9 KO cells stably expressing DiPAK were seeded onto poly-L-lysine–coated coverslips. One day after seeding, DiPAK expression was induced by the addition of 30 µg/ml cumate. For the last 16 h of DiPAK expression, cells were treated with 1 µl/ml DMSO, 10 µM 1G244, and 1 µM MG132, respectively. After the treatment/induction period, the media were removed, and cells were washed with 1 ml PBS and incubated with 1 ml fixation buffer (4% PFA in PBS; Thermo Fisher Scientific) for 15 min at RT. Then, cells were washed 3x with 1 ml PBS and incubated with 0.5 ml of 4′,6-diamidino-2-phenylindole (DAPI) solution (1 µg/ml; Roth) for 15 min at RT. After washing the cells with 1 ml PBS, coverslips were mounted on microscopy slides with Mowiol (Sigma-Aldrich) and DABCO (Roth) and dried for 1 day at 4°C. The cells were analysed by fluorescence microscopy.

## Fluorescence microscopy image acquisition

For image acquisition, an LSM 980 microscope with Airyscan 2 and multiplex (Carl Zeiss Microscopy) was used with a Plan-Apochromat 63x/1,4 Oil DIC objective and the GaAsP-PMT, Multi-Alkali-PMT detector. The cells were imaged at RT with oil as imaging medium. The following fluorochromes were used: GFP and mKate2. Images were displayed using the acquisition software ZEN 3.3 and were processed using the software OMERO.insight.

## Steady-state DiPAK measurements in HEK293 cells using a fluorescence microscopy reader

For steady-state measurements, $5 \times 10^3$ HEK293 WT or DPP9 KO cells stably expressing DiPAK variants were seeded onto poly-L-lysine–coated 96-well plates. 1 d after seeding, DiPAK expression was induced by the addition of 30 µg/ml cumate for 24 h. For the last 16 h of the induction period, cells were treated with 1 µl/ml DMSO, 10 µM 1G244, 10 µM sitagliptin, 10 µM KYP-2047, or 1 µM MG132, respectively. After incubation, media were removed and 100 µl minimal media (140 mM NaCl, 5 mM KCl, 1 mM MgCl$_2$, 2 mM CaCl$_2$, 20 mM Hepes, and 10 mM glucose) containing 10% FCS, 30 µg/ml cumate, and the respective treatments (1 µl/ml DMSO, 10 µM 1G244, 10 µM sitagliptin, 10 µM KYP-2047, 1 µM MG132) were added. For analysis of DPP9 overexpression, DPP9 variant expression was induced for the last 4 h of the DiPAK induction period by the addition of 1 µg/ml doxycycline. Subsequently, DiPAK fluorescence was analysed using a Cytation 3 fluorescence microscopy reader (Agilent, BioTek). The instrument was heated to 37°C and 5% CO$_2$. DiPAK mEGFP fluorescence and mKate2 fluorescence were measured at excitation/emission wavelengths of 469-17.5 nm/525-19.5 nm and 586-7.5 nm/647-28.5 nm, respectively. Steady-state fluorescence was measured by acquiring an image every 10 min for 60 min. The RRA (redox ratio analysis) program (Fricker, 2016), a semi-automated software package, was used to extract fluorescence intensities for both channels and each single cell. Downstream analysis was performed in R. First, single-cell values were averaged over the whole measurement period, and then, the mean of all single cells measured for one condition was calculated. The

DiPAK ratio was calculated by dividing the mEGFP signal by the mKate2 signal of each individual cell. For data visualization, a boxplot was created using ggplot2 that shows the median, and the 25th (lower hinge) and the 75th (upper hinge) percentile. The whiskers extend from the lower and upper hinge to the largest and smallest value, respectively, but no further than 1.5*IQR (interquartile distance) from the respective hinge. In addition, each cell is represented as an individual data point. Statistical comparison of different conditions was performed using Welch's *t* test.

### Time course DiPAK measurements in HEK293 cells using a fluorescence microscopy reader

For time course measurements, $5 \times 10^3$ HEK293 WT, DPP9 KO, or DPP9 KO + DPP9-S WT cells stably expressing DiPAK (and DPP9 KO + DPP9-S WT also stably expressing DPP9-S WT) were seeded onto poly-L-lysine–coated 96-well plates. One day after seeding, DiPAK expression was induced by the addition of 30 µg/ml cumate for 24 h. After incubation, media were removed and 50 µl minimal media containing 10% FCS and 30 µg/ml cumate were added. For 1G244 and doxycycline titration, 50 µl minimal media containing 1G244 or doxycycline were added to reach the final treatment concentrations indicated. 1 µl/ml DMSO was used as a control. Right after the addition of the treatments, DiPAK fluorescence was analysed over time using a Cytation 3 fluorescence microscopy reader (Agilent, BioTek) as described above. Images were acquired every 10 min for 4 h. The RRA (redox ratio analysis) program (Fricker, 2016), a semi-automated software package, was used to extract fluorescence intensities for both channels and each single cell. The DiPAK ratio was calculated by dividing the mEGFP signal by the mKate2 signal of each individual cell. Downstream analysis was performed in R. First, single-cell values were normalized to the first time point, and then, the mean of all single cells measured for one condition was calculated for each time point. For data visualization, a point plot was created using ggplot2 with each data point representing a single cell. The mean is displayed as a continuous line.

To better visualize changes in the DiPAK ratio, the DiPAK ratio of the DMSO control was subtracted from the DiPAK ratio of the 1G244/doxycycline-treated samples yielding the Δ(DiPAK ratio).

### DiPAK measurements in HEK293 cells by flow cytometry

For steady-state measurements, $5 \times 10^5$ HEK293 WT or DPP9 KO cells stably expressing DiPAK were seeded onto six-well plates. One day after seeding, DiPAK expression was induced by the addition of 30 µg/ml cumate for 24 h. For the last 16 h of the induction period, cells were treated with 1 µl/ml DMSO, 10 µM 1G244, or 1 µM MG132, respectively. After incubation, cells from one well were collected in 1 ml DMEM containing 10% FCS. Before analysis by flow cytometry, cells were stained with 1 µM SYTOX Blue Dead Cell Stain (Cat# S34857; Life Technologies). Then, DiPAK fluorescence was analysed using a BD FACSCanto II system (Becton Dickinson) and FlowJo V7.6.3 software (Treestar). Cells were gated in three steps. First, a population homogeneous in size and granularity was selected based on side scatter (SSC) and forward scatter (FSC). Then, viable cells were selected based on the SYTOX Blue dead cell stain, which was detected using the Pacific Blue channel. Finally, a broad gate was

set based on mEGFP and mKate2 fluorescence measured in the FITC and PerCP-Cy5.5 channels, respectively, to exclude widespread counts. The DiPAK ratio was calculated by dividing the mEGFP signal by the mKate2 signal of each individual cell. Downstream analysis was performed in R. For data visualization, a boxplot was created using ggplot2 that shows the median, and the 25th (lower hinge) and the 75th (upper hinge) percentile. The whiskers extend from the lower and upper hinge to the largest and smallest value, respectively, but no further than 1.5*IQR (interquartile distance) from the respective hinge. Statistical comparison of different conditions was performed using Welch's *t* test.

### Steady-state DiPAK measurements in HEK293 cells using a fluorescence plate reader

For steady-state measurements, $2 \times 10^6$ HEK293 WT or DPP9 KO cells stably expressing DiPAK were seeded onto 10-cm dishes. One day after seeding, DiPAK expression was induced by the addition of 30 µg/ml cumate for 24 h. For the last 16 h of the induction period, cells were treated with 1 µl/ml DMSO, 10 µM 1G244, or 1 µM MG132, respectively. After incubation, $4 \times 10^6$ cells were transferred to a Falcon tube and centrifuged for 2 min at 500*g* and RT. The supernatant was removed, and the cell pellet was resuspended in 2 ml warm PBS. Per technical replicate, 200 µl of the cell suspension (equal to $4 \times 10^5$ cells) was transferred to each well of a 96-well plate. Subsequently, the plate was centrifuged for 5 min at 20*g* and RT. Then, DiPAK fluorescence was analysed using a CLARIOstar (BMG) plate reader. The instrument was heated to 37°C. DiPAK mEGFP fluorescence and mKate2 fluorescence were measured at excitation/emission wavelengths of 469-17.5 nm/525-19.5 nm and 586-7.5 nm/647-28.5 nm, respectively. The DiPAK ratio was calculated by dividing the mEGFP signal by the mKate2 signal of each individual cell. Data were analysed using Microsoft Excel 2019.

### DiPAK measurements in melanoma cells

Human melanoma cell lines (WM1366, primary cell line; and WM3734a, brain metastatic cell line) were kindly provided by Prof. Meenhard Herlyn, the Wistar Institute, Philadelphia, USA. Cells were cultured in DMEM supplemented with 10% FCS and maintained at 37°C, 5% CO2. Both cell lines were regularly tested negative for *Mycoplasma* using PCR *Mycoplasma* Test Kit I/C (Cat# PK-CA91-1048; PromoCell, PromoKine).

WM1366 and WM3734a melanoma cells were seeded onto a 25-mm coverslip (Cat# 631-0172; VWR) and transiently transfected with 1 µg of DiPAK (with mScarlet-I in the normalizer unit) plasmid DNA using FuGENE HD and Opti-MEM (Cat# 51985034; Thermo Fisher Scientific) according to the manufacturer's instructions. Imaging was performed 48 h after transfection. Where applicable, cells were treated with DMSO or 10 µM 1G244 for the last 16 h of the transfection time before imaging.

Fluorescence microscopy measurements were performed in 0.5 mM Ca2+ Ringer buffer (145 mM NaCl, 4 mM KCl, 10 mM Hepes [4-(2-hydroxyethyl)-1-piperazineethanesulfonic acid, pH 7.4], 10 mM glucose, 2 mM MgCl2, and 0.5 mM CaCl2) at 37°C using a Zeiss D1 Cell Observer fluorescence microscope equipped with a 40x/1.3 oil Neofluar objective, Axiocam 702 mono, and LED system (Colibri;

Zeiss). Images were acquired upon excitation at 505 nm (500/15 nm, 20% LED) and 555 nm (550/32 nm, 20% LED), using 515- and 573-nm dichroic mirrors, and 539/25- and 630/92-nm emission filters. Single-cell data acquisition was performed using ZEN 2.6 software, and data processing and statistics with Microsoft Excel 2019 and GraphPad Prism 9. Data were plotted as a normalized steady-state 505/555-nm DiPAK ratio of a single cell.

## Isolation of murine B cells

Mice from different mouse lines (Kwon et al, 2008; Young et al, 2011; Baris et al, 2015) were anaesthetized using $CO_2$ and euthanized by cervical dislocation. The spleen was transferred into cold PBS containing 2% FCS. The spleen was minced and filtered through a 70-$\mu$M cell strainer (Falcon, #352350; Corning Life Sciences). Cell suspensions were transferred into 15-ml Falcon tubes and pelleted by centrifugation for 7 min at 470$g$ and 4°C. Erythrocytes were lysed in 5 ml red blood cell lysis buffer (#420301; BioLegend) at RT for 5 min. The reaction was stopped with 5 ml of cold PBS containing 2% FCS buffer. Cell suspensions were filtered through a 30-$\mu$m filter (#04-0042-2316; Sysmex) before centrifugation for 7 min at 470$g$ and 4°C. Cells were resuspended in PBS containing 2% FCS buffer, and the cell concentration was determined with the NucleoCounter NC-3000 (ChemoMetec) following the manufacturer's protocol. B cells from splenic single-cell suspensions were magnetically enriched using EasySep Mouse B-cell Isolation Kit (#19854; StemCell) following the manufacturer's specifications. The quality of the B-cell purification was assessed by flow cytometry using CD19 and B220 antibodies for surface staining. B-cell enrichment after isolation was consistently above 95%.

## In vitro cultivation of primary murine B cells

Isolated splenic B cells (0.5 × $10^6$ cells/ml) were cultured in complete R10 medium (RPMI 1640 [Cat# 31870-25; Thermo Fisher Scientific], 10% FCS, 2 mM glutamate, 1 mM sodium pyruvate, 50 U/ml penicillin G, 50 $\mu$g/ml streptomycin, 50 $\mu$M $\beta$-mercaptoethanol) at 37°C and 5% CO2 for 72 h with the addition of 10 $\mu$g/ml LPS (Cat# L3024; Sigma-Aldrich) as previously described (Urbanczyk et al, 2022).

## Retroviral infection of primary murine B cells

The cDNA encoding the DiPAK sensor was cloned into the retroviral vector pCru5 (Steinmetz et al, 2021) by the Gibson assembly. Supernatants containing retroviruses were obtained by transfection of the retroviral construct in Platinum-E cells. Then, 0.5 × $10^6$ splenic B cells activated with LPS were spin-infected with 2 ml of retroviral supernatants and 4 $\mu$g/ml Polybrene for 3 h at 33°C and 1,480$g$. Cells were washed in medium and cultivated further.

## Extracellular flux analysis (Seahorse)

Seahorse experiments were performed as described previously (Urbanczyk et al, 2022). Briefly, the day before the experiments, cell plates were coated with 10 $\mu$g/ml poly-L-lysine in 0.01 M Tris–EDTA buffer, pH 8.0. Splenic naïve B cells were isolated and activated with

LPS in vitro. After 3 d, LPS blasts were seeded at a density of 2.5 × $10^5$ cells/well and measured at least in triplicates. Extracellular flux analysis was performed as described previously (Urbanczyk et al, 2022). Mito Stress Test and calculations were performed following the guidelines provided by the manufacturer (Agilent) and using WAVE software.

## Cell cycle and apoptosis analysis of murine cells

To analyse the cell cycle status, seven cultures of 2 × $10^6$ B cells in 4 ml R10 medium were prepared, induced with LPS, and inhibited with different 1G244 concentrations (2, 4, 6, 8, 10, and 12 $\mu$M). One culture was left untreated as a control group. On day 2, 1 ml of each condition was transferred in FACS tubes, pelleted, resuspended in 100 $\mu$l propidium iodide solution (50 $\mu$g/ml propidium iodide, 0.1% sodium citrate, 0.1% Triton X-100), and incubated for at least 2 h at 4°C in the dark (Nicoletti et al, 1991). Apoptosis was analysed by staining the cells with Apotracker Green (#427403; BioLegend) in PBS, followed by Fixable Viability Dye eFluor 780 (#65-0865-14; eBioscience). All samples were measured using a Beckman Coulter Gallios flow cytometer.

## Flow cytometry of murine cells

For the flow cytometric analysis, the cell number was adjusted to 0.5–2 × $10^6$ cells per FACS tube in a volume of 500 $\mu$l per sample. For the murine cells, before labelling surface proteins, non-specific antibody binding was blocked by using a 1:100 dilution of unlabelled anti-CD16/CD32 (Cat# 14-0161-86; Invitrogen) in FACS buffer (consisting of PBS containing 2% FCS) for 15 min at 4°C. Cells were washed with FACS buffer and stained for 15 min at 4°C with the appropriate amount of the following fluorophore-conjugated antibodies: TACI/CD267 APC (1:400), CD138 PE-Cy7 (1:1,500), B220/CD45R PerCP-Cy5.5 (1:200), CD19 APC-Fire750 (1:200). A representative gating strategy is shown in Fig S7.

## DiPAK measurements in murine cells by flow cytometry

For analysis of DiPAK signals, primary murine B cells were first induced with LPS. The next day, cells were infected with the DiPAK sensor and cultured for 36 h. Then, one culture was treated with 8 $\mu$M 1G244, whereas another culture was left untreated. Subsequently, cells were analysed by flow cytometry. Next to the markers described above, DiPAK mEGFP fluorescence and mKate2 fluorescence were detected using the GFP FITC and mCherry ECD channels, respectively. The DiPAK ratio was calculated by dividing the mEGFP signal by the mKate2 signal of each individual cell. Downstream analysis was performed in R. For data visualization, a scatter plot was created using ggplot2 showing the signals for the CD138 and TACI surface markers on the x- and y-axis, respectively. For mapping the DiPAK ratio on the scatter plot, the plot area was divided into 10,000 bins. For each bin, the mean DiPAK ratio of all data points (representing a single cell each) within this bin was calculated. The bins were then coloured according to their mean DiPAK ratio. This combined approach allows distinguishing activated B cells and plasmablasts based on the TACI and CD138 signals, while simultaneously assessing the DiPAK ratio.

To further visualize and compare the DiPAK ratios of activated B cells and plasmablasts, boxplots were created using ggplot2. These show the median, and the 25th (lower hinge) and the 75th (upper hinge) percentile. The whiskers extend from the lower and upper hinge to the largest and smallest value, respectively, but no further than 1.5*IQR (interquartile distance) from the respective hinge. Statistical comparison of different conditions was performed using Welch's *t* test.

### Quantification and statistical analysis

The number of experiments/cells analysed is reported in the figure legend. Details on statistical analyses are provided in the respective method sections and figure legends.

### Online supplemental material

Fig S1 shows how the GP-AMC DPP activity assay and AK2(1–15)-mEGFP-Strep can function as readouts for DPP8/9 activity. In addition, data on HEK293 cell viability upon DPP8/9 inhibition are provided. Fig S2 demonstrates that DiPAK expression does not affect cell growth and holds data for a DiPAK measurement in HeLa Flp-In T-Rex cells. Fig S3 shows a comparison of the mean DiPAK ratio fold changes calculated based on the measurements shown in Fig 3B–D and provides an analysis of the effect of DPP4 and PREP inhibition on the DPP activity and DiPAK ratio in HEK293 cells. Fig S4 summarizes the characterization of DiPAK variants with different normalizer moieties. Fig S5 holds single-channel data for the time-resolved DiPAK measurements shown in Fig 4 and an assessment of DPP8/9 activity in lysates of cells with different DPP9 protein levels. In addition, data on DiPAK measurements upon the overexpression of DPP9-S and DPP9-L in HEK293 cells are provided. Fig S6 displays survival analysis of cancer patients with high and low DPP9 expression levels. Furthermore, DPP8 and DPP9 levels and the influence of DPP8/9 on WM1366 and WM3734a viability were analysed. Fig S7 shows cell cycle analysis of LPS-activated B cells upon DPP8/9 inhibition, statistical analysis of oxygen consumption rate assessment of LPS-activated B cells upon DPP8/9 inhibition, and a representative pregating strategy for flow cytometric analysis of activated B cells. Fig S8 holds a characterization of B-cell viability upon DPP8/9 and caspase-1 inhibition. Fig S9 shows how the number of activated B cells and plasmablasts/plasma cells is affected upon DPP8/9 and caspase-1 inhibition. Table S1 lists the properties of fluorescent proteins used in the DiPAK variants. Table S2 contains the cell lines used in this study. Table S3 lists plasmids and primers used in this study. Table S4 contains the antibodies used in this study. Table S5 contains chemicals and further tools used in this study. Table S6 lists experimental models used in this study. Table S7 lists software and algorithms employed in this study.

## Data Availability

All materials described here and in the Supplemental Data are available upon reasonable request from the corresponding author.

Raw data are included as dataset files in the Supplementary Data section of the article.

## Supplementary Information

## Acknowledgements

The Deutsche Forschungsgemeinschaft (DFG, German Research Foundation) funds research in the Laboratory of JR through the grants RI2150/5-1, project number 435235019, RTG2550/2, project number 411422114, SPP2453, project number 541742459, CRC1218, project number 269925409, and CRC1678, project number 520471345. D Mielenz is supported by the Deutsche Forschungsgemeinschaft (DFG, German Research Foundation) through the grants FOR5560/Mi939/7-1, project number 505372148, and Mi939/6-1, project number 503852185. Research of I Bogeski is funded by the Deutsche Forschungsgemeinschaft (DFG, German Research Foundation) through the grants CRC1190, project number 200049484, and BO 3643/10-1, project number 545970076. B Brachvogel is supported by the DFG grant FOR2722, BR2304/12-1, project number 407146744, and BR2304/7-1, project number 207342459. J Etich is supported by the DFG grant FOR2722. We thank Kathrin Ulrich and members of the Riemer laboratory for critical reading of the article. We thank Anja Wittmann and Anika Seiler for technical support throughout the project. We thank the CECAD Imaging Facility for their support in microscopy. We thank Christian Ickes for his assistance with the melanoma survival analyses.

### Author Contributions

K Weiss: conceptualization, resources, data curation, formal analysis, validation, investigation, visualization, methodology, and writing—original draft, review, and editing.
Y Agarkova: formal analysis, investigation, and methodology.
A Zwosta: formal analysis, investigation, and methodology.
S Hoevel: formal analysis, investigation, and methodology.
A-K Himmelreich: formal analysis, investigation, and methodology.
M Shumanska: formal analysis, investigation, methodology, and writing—review and editing.
J Etich: formal analysis, investigation, and methodology.
G Poschmann: formal analysis, investigation, and methodology.
B Brachvogel: formal analysis, supervision, funding acquisition, and writing—review and editing.
I Bogeski: formal analysis, supervision, funding acquisition, and writing—review and editing.
D Mielenz: formal analysis, supervision, funding acquisition, and writing—review and editing.
J Riemer: conceptualization, resources, data curation, formal analysis, supervision, funding acquisition, validation, visualization, project administration, and writing—original draft, review, and editing.

### Conflict of Interest Statement

The authors declare that they have no conflict of interest.

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
