## [Reviewer comments · Life Science Alliance]

Life Science Alliance

A fluorescent sensor for real-time monitoring of DPP8/9 reveals crucial roles in immunity and cancer

Konstantin Weiss, Yelizaveta Agarkova, Alexandra Zwosta, Sarah Hoevel, Ann-Kathrin Himmelreich, Magdalena Shumanska, Julia Etich, Gereon Poschmann, Bent Brachvogel, Ivan Bogeski, Dirk Mielenz, and Jan Riemer

DOI: <https://doi.org/10.26508/lsa.202403076>

Corresponding author(s): Jan Riemer, University of Cologne

Review Timeline:	Submission Date:	2024-10-07
	Editorial Decision:	2024-11-04
	Revision Received:	2025-03-28
	Editorial Decision:	2025-04-18
	Revision Received:	2025-04-30
	Accepted:	2025-04-30

Scientific Editor: Tim Fessenden

Transaction Report:

November 4, 2024

Re: Life Science Alliance manuscript #LSA-2024-03076-T

Prof. Jan Riemer
University of Cologne
Biochemistry
Zuelpicher Str. 47a
Cologne 50674
Germany

Dear Dr. Riemer,

Thank you for submitting your manuscript entitled "A fluorescent sensor for real-time monitoring of DPP8/9 reveals crucial roles in immunity and cancer" to Life Science Alliance. The manuscript was assessed by expert reviewers, whose comments are appended to this letter. We invite you to submit a revised manuscript addressing the Reviewer comments.

Thank you for this interesting contribution to Life Science Alliance. We are looking forward to receiving your revised manuscript.

Sincerely,

B. MANUSCRIPT ORGANIZATION AND FORMATTING:

Reviewer #1 (Comments to the Authors (Required)):

This ms contains excellent biochemistry presenting a new biological reagent, termed DiPAK, that was designed to measure intracytoplasmic catalytic activity derived from the two similar proteases DPP8 and DPP9. The ability to measure changes in DPP8/9 catalytic activity in real time is shown and is an exciting advance for the field.

However, this ms needs to become very clear on these two topics:

1. whether DiPAK also detects/measures DPP4 or PREP activity to some extent. This may be more relevant when cell types, unlike HEK293, that are high in PREP or DPP4 are used in future. For point 1, perhaps the previous papers on DPP9 cutting AK2 looked at whether DPP4 or PREP can also cut AK2. In addition, a key experiment that used 1g244 should be run again with 1g244 compared to a DPP4-selective inhibitor such a sitagliptin and a PREP-selective inhibitor. Currently, it is assumed that the residual activity in 9-KO cells is DPP8, but data is needed .

2. Whether it detects DPP9 that is in the nucleus. The two main DPP9 forms, DPP9 short and DPP9 long are expressed in ~equal amounts and traffic to cytoplasm and nucleus, respectively. It appears that DPP8/9 activity in nucleus is not detected by DiPAK; please clarify and specify this limitation. For point 2, modifying the text might be sufficient, but a possible experiment would be to transfect in the DPP9 long form and then add various inhibitors.

The term DPP8/9 might be questioned by some readers, but 'DPP8/9' is already in the literature as meaning DPP8 and DPP9 together.

In the literature, 1g244 maximally inhibits DPP9 and 8 by about half; that fits with data in this ms.

The title mentions cancer, but no data in this ms is on cancer. Reconsider the title. It would be helpful to show that DiPAK works in another cell line(s), such as HeLa.

Reviewer #2 (Comments to the Authors (Required)):

Summary of major findings

DPP8 and DPP9 are intracellular serine proteases involved in various physiological processes. To assist the further assessment of these proteases in human cells, methods to examine their activity in living cells would be valuable tools for this research field. The authors developed DiPAK as a method to detect DPP8/9 activity in cells via their ability to process the N-terminus of adenylate kinase 2 (AK2). The system can be applied for a wide range of applications (Western blotting, fluorescence microscopy, flow cytometry, plate-based fluorescent assays) and seems suitable for application by other researchers. The introduction contains a summary of the state-of-the-art concerning DPP8/9 activity in human cells with appropriate references. However, no notion is made about their role in lytic cell death by interacting with inflammasome-forming proteins, which currently constitutes a major part of DPP8/9 literature and research interest.

Major remarks about the experimental design and results:

General: Quite high doses of 1G244 are used; Interpretation of results will be facilitated when the effect on cell viability is addressed at the concentrations and time intervals used throughout the study. These experimental data should be added.

- Results in Fig 1C,D,E,H: why was proteasome inhibition not included as a control? The authors show that both, DPP9 inhibition and 3CS mutation stabilize AK2 levels, but not that AK2 is directed to the proteasome in their model when DPP8/9 are enzymatically active. Blocking the proteasome before AK2/AK2 variant expression should stabilize their levels.

- Results in Fig 2C: what are the DPP8 levels in DPP9 KO cells?

- DPP8/9 inhibition and DPP9 deficiency are reported to induce lytic cell death. Did you assess the effects of DPP9 KO, 1G244 (see above) or MG132 treatment on cell viability in your models? This is necessary information for a proper interpretation of the results.

- Results in Figure 3A: please include (in suppl. data) the negative control staining of cells without DiPAK expression acquired

under the same conditions as the DiPAK-expression cells.

- Results in Figure 4:

o Already after 1h 1G244, the effect on AK2 processing is visible. In the previous experiments, the read-out was after 16h. Why was the effect not followed up to 16h in Fig 4? 1G244 has no effect in DPP9 KO cells after 4h, while after 16h there was some effect (cfr. Fig 2) on the DiPAK ratio; this could be addressed in the discussion of the manuscript.

- The authors use WM1366 and WM3734a cells. To the best of my knowledge, these cells have not yet been used in DPP8/9 research and it is unknown whether they are sensitive to DPP8/9 inhibition-induced cell death. The authors show that DPP9 protein levels differ between the two cell types. It would be interesting to also assess DPP8 protein levels e.g. via Western blot and the DPP8/9 activity via the GP-AMC assay in the two cell types in order to allow a proper discussion of the difference in DiPAK ratio.

- Figure 6B: Please add information on how cell growth was measured. Please adapt the graph to make it more clear.

- Please add full Western blot membranes to the supplementary file.

- General remarks for figures:

o Please add units to axes in graphs where necessary.

o Please add the number of independent repeats for each experiment (e.g. missing in legend of Fig 3A)

Minor textual remarks about the abstract, significance statement and introduction:

- Line 37: Please add what 'DiPAK' stands for.

- Lines 46-49: 'In hundreds of proteins, the dipeptidyl peptidases 8 and 9 (DPP8/9) remove two amino acids if the second is proline, affecting protein half-life, protein translocation, and protein-protein interactions.'. This has been predicted and is probably true but has not yet been shown in cells. Only for a number of substrates, the physiological relevance has been shown so far. Please adapt in the manuscript.

- Lines 137-138: please add reference for the cleavage of AK2 by MAP.

The authors combined the use of cell lines and primary cells, which is highly valuable and relevant.

The manuscript would benefit from a short discussion on possible species differences. Now human melanoma cell lines are used and primary murine B cells without addressing this point.

Materials and methods

- Please add commercial sources for 1G244 and MG132.

Discussion

- How do we know that AK2 processing by DPP8/9 is a reliable readout for DPP8/9 activity? Please discuss the possible processing of AK2 N-terminus by other intracellular proteases. - In cells with DPP9 bound to NLRP1, which inserts its C-terminus in the active site of the protease, it would probably not be able to process AK2 or the sensor. DPP8/9 interacts with proteins in a manner dependent on cell type and activation status. The effect of interactions on its in situ enzymatic activity or on its interactions with the sensor are hard to predict.

In this regard, it seems a bold statement that DiPAKs might have a future as tools to evaluate the nature of cancer cells (lines 376-379).

- The authors do not mention that DPP8/9 inhibition with 1G244 can induce cell death via the inflammasome, which might influence the interpretation of many results in this manuscript. Please discuss the available literature on the effects of DPP8/9 inhibition on cell viability in the different experiments of the manuscript.

Reviewer #3 (Comments to the Authors (Required)):

Summary:

The authors developed a genetically encoded fluorescent DPP8/9 activity sensor that can be used for in-cell activity measurement of these two proteases. For the development of this sensor, they exploited a previously identified DPP9 substrate, AK2. Briefly, N-terminal processing of AK2 by methionine aminopeptidases reveals the DPP9 cleavage site (an Ala-Pro N-terminus). Removal of the AP dipeptide exposes an unstable N-terminal residue, which in turn leads to AK2 proteasomal degradation by the N-end rule pathway. By fusing the first 15 residues of AK2 to GFP (AK2 1-15 GFP), they can monitor DPP8/9 activity by measuring GFP fluorescence (ie inhibition of DPP8/9 gives greater fluorescence). As a control, they performed a single point mutation on the AK2 N-terminus, S4P, (AK2 1-15 S4P GFP), which prevents DPP8/9 cleavage. By combining these two ORFs into one plasmid, separated by an IRES site they can monitor DPP8/9 activity by measuring the stability of the GFP reporter without being confounded by expression levels of the reporter. The authors provide appropriate controls for this reporter and verify that it is DPP8/9 specific. They show that their reporter can be stabilized by 1G244, a DPP8/9 inhibitor which prevents AP processing and by MG132, a proteasome inhibitor, which prevents GFP degradation following the exposure of the N-degron. Importantly, the fluorescence of this reporter can be assessed by three independent methods: by fluorescence microscopy, flow cytometry and fluorescence plate reader, allowing its application in multiple cell lines and experiments set up. Overall, this is an well-designed and useful DPP8/9 activity in-cell reporter. However, the application of the reporter in Figure 6 has major issues, as all of the assays are could simply be observing DPP8/9 induced pyroptosis (ie cell death confounding their interpretation) .

Major comments:

Figure 6: DPP8/9 inhibition can induce pyroptosis in resting lymphocytes, including T cells and B cells (Johnson, et al. 2020, Linder et al. 2020). It seems likely that 1G244 is inducing cell death in Fig 6B-D, thereby explaining their phenotypic responses. The authors should:

1. Include cell viability assays (Cell-titer GLO or LDH release assays) for the cells treated with the different concentrations of 1G244. This is not only important for controlling for DPP8/9 inhibitor-induced pyroptosis, but also for controlling for 1G244 off-target-mediated cell death (Okondo et al, 2017).
2. Perform the same experiments in the presence of a caspase-1 inhibitor (VX-765).
3. Figure 6B and S5A: It would be nice to include a DiPAK ratio measurement. Does the decrease in B cell growth correlate with DPP8/9 activity as measured by the DiPAK ratio in the presence of a caspase-1 inhibitor?
4. Figure 6E: it seems like there really isn't a DPP8/9 activity difference in activated b cells and plasmablasts. Moreover, it seems like there shouldn't really be any DPP8/9 activity at 8 uM 1G244. Is this figure scaled to make a very minor difference seem large?
5. 1G244 has DPP8/9 inhibition-independent effects at concentrations >10 uM (Okondo et al, 2017), which the authors use in Figure 6. The authors should use one completely unrelated DPP8/9 inhibitor (VbP) to show that the effect is really due to DPP8/9 inhibition.

Minor comments:

Figure 3A: would be nice to include the brightfield versions of the respective fluorescence microscopy images.

Weiss et al; Point-by-point response to the reviewer comments**Reviewer #1 (Comments to the Authors (Required)):**

This ms contains excellent biochemistry presenting a new biological reagent, termed DiPAK, that was designed to measure intracytoplasmic catalytic activity derived from the two similar proteases DPP8 and DPP9. The ability to measure changes in DPP8/9 catalytic activity in real time is shown and is an exciting advance for the field.

However, this ms needs to become very clear on these two topics:

1. whether DiPAK also detects/measures DPP4 or PREP activity to some extent. This may be more relevant when cell types, unlike HEK293, that are high in PREP or DPP4 are used in future. For point 1, perhaps the previous papers on DPP9 cutting AK2 looked at whether DPP4 or PREP can also cut AK2. In addition, a key experiment that used 1g244 should be run again with 1g244 compared to a DPP4-selective inhibitor such a sitagliptin and a PREP-selective inhibitor. Currently, it is assumed that the residual activity in 9-KO cells is DPP8, but data is needed.

We thank the reviewer for pointing out potential applications of DiPAK in cell lines that are high in DPP4 or PREP activity. To address whether DiPAK also measures DPP4 and PREP activity in HEK293 cells, we compared changes in the DiPAK ratio upon treatments with 1G244 (DPP8/9 inhibition), Sitagliptin (DPP4 inhibition), KYP-2047 (PREP inhibition) and MG132 (proteasomal inhibition) (**new Figure S3B,C**). This experiment shows that the DiPAK ratio is only increased upon treatment with the DPP8/9 inhibitor 1G244 as well as upon proteasomal inhibition, but not upon treatment with the DPP4 or PREP inhibitors. As we observe the same behavior of the DiPAK ratio in DPP9 KO cells, we conclude that remaining DPP activity measured is not based on DPP4 or PREP but most likely based on DPP8.

To further verify that DPP activity in our HEK293 cell model is not based on DPP4 or PREP, we conducted a GP-AMC assay to measure DPP activity in HEK293 Flp-In T-Rex Wt and DPP9 KO cell lysates (**new Figure S3D**). We could demonstrate that DPP activity in both, WT and DPP9 KO lysates, is only decreased upon addition of the DPP8/9 inhibitor but not upon addition of Sitagliptin or KYP-2047. This again confirms, that DPP activity in HEK293 Flp- In T-Rex WT cells as well as remaining DPP activity in DPP9 KO cells is not based on DPP4 or PREP.

2. Whether it detects DPP9 that is in the nucleus. The two main DPP9 forms, DPP9 short and DPP9 long are expressed in ~equal amounts and traffic to cytoplasm and nucleus, respectively. It appears that DPP8/9 activity in nucleus is not detected by DiPAK; please clarify and specify this limitation. For point 2, modifying the text might be sufficient, but a possible experiment would be to transfect in the DPP9 long form and then add various inhibitors.

We thank this reviewer for highlighting that DiPAK measurements can possibly employed in a compartment-specific manner. DiPAK is localized to the cytosol (**Figure 3A**), where it is accessible for DPP9-S processing, but presumably not accessible for processing by the nuclear encoded DPP9 variant

DPP9-L. To determine whether, the DiPAK ratio is affected by DPP9-L activity, we expressed DPP9-S and DPP9-L, respectively, in HEK293 Flp-In T-Rex DPP9 KO cells. We found that overexpression of both, DPP9-S and DPP9-L, resulted in a decrease in the DiPAK ratio, which can be counteracted by treating the cells with 1G244 or MG132 in parallel (**new Figure S5D,E**). These findings indicate, that DiPAK is capable of detecting DPP9-L activity. It is unclear, however, whether the activity detected is based on DPP9-L, which is localized to the nucleus or cytosolic DPP9-L as Justa-Schuch et al., 2014 found that while DPP9-L is highly enriched in the nuclear fraction of HeLa cell lysates, also a small portion is present in the cytosol. We could imagine, that upon overexpression, a significant fraction of DPP9-L might also exhibit cytosolic localization or based on NLS trafficking, is shuttling back and forth between the cytosol and the nucleus. Further, considering that the change in the DiPAK ratio upon DPP9-S and DPP9-L overexpression is relatively small compared to the changes upon DPP8/9 inhibition (**Figure 3, 4**), it is likely, that already a small portion of DPP9-L present in the nucleus could significantly change the DiPAK ratio as observed in Figure S5D and E.

Additionally, in the HEK293 Flp-In T-Rex DPP9 KO cells used, DPP8 is still present. To our knowledge, it has not been investigated so far, if DPP8 and DPP9-L are capable of forming heterodimers, which then could then result in increased DPP8/9 activity in the cytosol upon DPP9-L overexpression.

The term DPP8/9 might be questioned by some readers, but 'DPP8/9' is already in the literature as meaning DPP8 and DPP9 together. In the literature, 1g244 maximally inhibits DPP9 and 8 by about half; that fits with data in this ms. The title mentions cancer, but no data in this ms is on cancer. Reconsider the title.

In **Figure 5** and accompanying supplementary figures, we analyse melanoma cells derived from skin cancer (**Figure 5 and S6**). Upon request of reviewer #2, we also extended this data set (see below).

It would be helpful to show that DiPAK works in another cell line(s), such as HeLa.

In the previous version of the manuscript, we already provided data on DiPAK functionality in HEK293 cells (**Figure 1-4**), in a primary and a metastatic melanoma/ skin cancer cell line (WM1366 and WM3734a, respectively, **Figure 5**), and in primary B cells derived from mouse spleen (also upon differentiation, **Figure 6**). As requested we also conducted a DiPAK measurement in HeLa Flp-In T-Rex cells, using Western blot as a readout (**new Figure S2B**).

Reviewer #2 (Comments to the Authors (Required)):

Summary of major findings

DPP8 and DPP9 are intracellular serine proteases involved in various physiological processes. To assist the further assessment of these proteases in human cells, methods to examine their activity in living cells would be valuable tools for this research field. The authors developed DiPAK as a method to detect DPP8/9 activity in cells via their ability to process the N-terminus of adenylate kinase 2 (AK2). The system can be applied for a wide range of applications (Western blotting, fluorescence microscopy, flow cytometry, plate-based fluorescent assays) and seems suitable for application by other researchers.

The introduction contains a summary of the state-of-the-art concerning DPP8/9 activity in human cells with appropriate references. However, no notion is made about their role in lytic cell death by interacting with inflammasome-forming proteins, which currently constitutes a major part of DPP8/9 literature and research interest.

In the introduction, we extended the presentation of DPP9 functions by pyroptosis, and we also further added the capacity of DPP9 to bind to NLRP1 and CARD8 in controlling inflammasome activation.

Major remarks about the experimental design and results:

General: Quite high doses of 1G244 are used; Interpretation of results will be facilitated when the effect on cell viability is addressed at the concentrations and time intervals used throughout the study. These experimental data should be added.

We assessed cell viability over time upon treatment with different amounts of 1G244. Viability of HEK293 cells treated with up to 10 μ M 1G244 remained comparable to untreated cells for at least 96 hours (**new Figure S1B**). Likewise, cell proliferation of HEK293 cells was not affected by 1G244 treatment, and also did not differ between WT and DPP9 KO cells (**new Figure S2A**).

We also tested susceptibility to 1G244 for some of the other cell lines employed in this study: WM1366 cells were not affected upon treatment with 10 μ M 1G244 for up to 72 hours and WM3734a cells were only slightly affected under these conditions (**new Figure S6C**).

Conversely, B cells require DPP8/9 activity for maturation and are sensitive to DPP8/9 loss also when unstimulated (**new Figures S8, S9**). The viability and cell death modalities found in stimulated and unstimulated B-cells are now extensively characterized in the manuscript (**Figure 6, Figure S7, new Figures S8-S9; see also answer to comment from referee #3 below**).

- Results in Fig 1C,D,E,H: why was proteasome inhibition not included as a control? The authors show that both, DPP9 inhibition and 3CS mutation stabilize AK2 levels, but not that AK2 is directed to the proteasome in their model when DPP8/9 are enzymatically active. Blocking the proteasome before AK2/AK2 variant expression should stabilize their levels.

We characterized the AK2-dependency on proteasomal degradation extensively in **Finger and Habich et al., 2020, EMBO**. Indeed, cytosolic AK2 is stabilized upon proteasome inhibition.

- Results in Fig 2C: what are the DPP8 levels in DPP9 KO cells?

We analyzed DPP8 levels in HEK293 Flp-In T-Rex DPP9 KO cells by mass spectrometry and found them to be not significantly changed when comparing protein levels in HEK293 Flp-In T-Rex DPP9 WT and DPP9 KO cells (**new Figure S1C**). DPP9 was not detected in DPP9 KO cells anymore.

- DPP8/9 inhibition and DPP9 deficiency are reported to induce lytic cell death. Did you assess the effects of DPP9 KO, 1G244 (see above) or MG132 treatment on cell viability in your models? This is necessary information for a proper interpretation of the results.

We tested cell viability and proliferation as described above for 1G244 treatment regimens and the DPP9 KO. We also tested the effects of MG132 on cell viability. Not unexpectedly, extended treatment of cells with MG132 killed them.

[Figure removed by editorial staff per authors' request]

- Results in Figure 3A: please include (in suppl. data) the negative control staining of cells without DiPAK expression acquired under the same conditions as the DiPAK-expression cells.

We repeated the experiment in Figure 3A including controls without DiPAK expression and replaced the former panel by this dataset (**Figure 3A**). Additionally, we provide in this point-by-point response also the brightfield images.

- Results in Figure 4: Already after 1h 1G244, the effect on AK2 processing is visible. In the previous experiments, the read-out was after 16h. Why was the effect not followed up to 16h in Fig 4? 1G244 has no effect in DPP9 KO cells after 4h, while after 16h there was some effect (cfr. Fig 2) on the DiPAK ratio; this could be addressed in the discussion of the manuscript.

We decided to perform this dynamic assessment of DPP9 overexpression only for up to 4 hours as the measurable increase in DiPAK signal in HEK293 cells upon 1G244 treatment takes mainly place in this time frame and then later a plateau is reached (*see figure below*).

There is very little residual DPP activity in DPP9 KO cells. Thus, we do not expect large differences between 1G244-treated and untreated cells in the sort-term. Still already after 4 hours of 1G244 treatment in DPP9 KO cells, we already do observe small increases in DiPAK signal compared to the

DMSO control (Figure 4A) indicative for this residual DPP activity. This increases upon longer 1G244 treatments as we perform them in other experimental settings in this manuscript (see e.g. Figure 3, 16 hr of 1G244)

- The authors use WM1366 and WM3734a cells. To the best of my knowledge, these cells have not yet been used in DPP8/9 research and it is unknown whether they are sensitive to DPP8/9 inhibition-induced cell death. The authors show that DPP9 protein levels differ between the two cell types. It would be interesting to also assess DPP8 protein levels e.g. via Western blot and the DPP8/9 activity via the GP-AMC assay in the two cell types in order to allow a proper discussion of the difference in DiPAK ratio.

Also, to our knowledge, DPP8/9 activity and function has not been characterized in melanoma before. We included melanoma cells as a further cellular model to demonstrate the applicability of the DiPAK sensor.

As requested by this referee we extended the characterization of these cells beyond the correlation between DPP levels and patient survival (**Figure S6A**), the proteomic assessment of DPP9 levels (**Figure 5**) and DiPAK response (**Figure 5**). We determined DPP9 and DPP8 levels by Western Blot (**new Figure S6B**), and performed viability assays upon DPP8/9 inhibition (**new Figure S6C**).

- Figure 6B: Please add information on how cell growth was measured. Please adapt the graph to make it more clear.

We added to the figure legend that cells were counted each day and counterstained with trypan blue to distinguish live and dead cells.

- Please add full Western blot membranes to the supplementary file.

The full Western Blot membranes are included in the source data for the respective figures.

- General remarks for figures:

o Please add units to axes in graphs where necessary.

We carefully checked the graph axes and added units where they were missing (e.g. in Figure 3 and 6). DiPAK ratio does not have a unit.

o Please add the number of independent repeats for each experiment (e.g. missing in legend of Fig 3A)

We included the number of replicates for each experiment in the respective figure legends. In case of Figure 3A, we show representative images of experiments that were performed in up to three biological replicates.

Minor textual remarks about the abstract, significance statement and introduction:
- Line 37: Please add what 'DiPAK' stands for.

Done.

- Lines 46-49: 'In hundreds of proteins, the dipeptidyl peptidases 8 and 9 (DPP8/9) remove two amino acids if the second is proline, affecting protein half-life, protein translocation, and protein-protein interactions.'. This has been predicted and is probably true but has not yet been shown in cells. Only for a number of substrates, the physiological relevance has been shown so far. Please adapt in the manuscript.

We adapted the text to include the word "predict".

- Lines 137-138: please add reference for the cleavage of AK2 by MAP.

We added the reference "Finger and Habich, EMBO, 2020"

The authors combined the use of cell lines and primary cells, which is highly valuable and relevant. The manuscript would benefit from a short discussion on possible species differences. Now human melanoma cell lines are used and primary murine B cells without addressing this point.

DPP9s from mouse and human have more than 90% sequence conservation. In both species AK2 is a target of DPP8/9 with the same cleavage site "MAP-". For other substrates of DPP8/9 the cleavage sites and the subsequent recognition mechanisms of the neo-N-termini might vary. However, in both species, DiPAK is functional and can serve as a valuable tool to detect DPP8/9 activity. We now emphasize in the discussion that we analysed human melanoma cells and primary murine B-cells but refrained from a more extensive discussion about species differences.

Materials and methods

- Please add commercial sources for 1G244 and MG132.

We added the commercial sources for 1G244 and MG132. They can be found in **Table S5**.

Discussion

- How do we know that AK2 processing by DPP8/9 is a reliable readout for DPP8/9 activity? Please discuss the possible processing of AK2 N-terminus by other intracellular proteases.

In our previous study on competition of mitochondrial protein import with DPP8/9-induced proteasomal degradation, we demonstrate that AK2 is sensitized for proteasomal degradation by DPP8/9 processing (Finger and Habich et al., EMBO, 2020). We verified these findings in Figure 1 of this manuscript and could confirm that mutation of the DPP8/9 cleavage motif at the N-Terminus of AK2 is resulting in stabilization.

To extend the reliability of these findings and exclude that further DPP family members with recognition motifs similar to DPP8/9 mediate AK2 stability, we compared the effects of DPP8/9, DPP4 and PREP inhibition using DiPAK as well as an in vitro GP-AMC activity assay. We found that DPP activity in HEK293 cells measured using both methods is not based on DPP4 or PREP, but rather DPP8/9 (**new Figure S3B-D**). Further, we added DiPAK measurements upon DPP9 overexpression, which resulted in a decreased DiPAK indicating a direct dependency of DiPAK on DPP9 activity (**new Figure S5D,E**). Additionally, the DiPAK ratio correlates with DPP9 levels not only in HEK293 cells (comparing WT vs DPP9 KO), but also in the melanoma cell lines WM1366 and WM3734a (**Figure 5**).

We mention these further experiments in the results and discussion section.

- In cells with DPP9 bound to NLRP1, which inserts its C-terminus in the active site of the protease, it would probably not be able to process AK2 or the sensor. DPP8/9 interacts with proteins in a manner dependent on cell type and activation status. The effect of interactions on its in situ enzymatic activity or on its interactions with the sensor are hard to predict. In this regard, it seems a bold statement that DiPAKs might have a future as tools to evaluate the nature of cancer cells (lines 376-379).

The regulation of catalytic and non-catalytic roles of DPP8/9 is a highly interesting topic and yet to be further investigated as it is unclear, how different shares of the cellular DPP8/9 pool are assigned with these roles. Thus, we consider and also discuss DiPAK as a valuable tool, to assess the levels of DPP8/9 processing activity, which are not involved in other processes. As this reviewer already pointed out, the available DPP8/9 processing capacity differs based on cell types and activation status, and, therefore, could serve as a readout for evaluating cancer cell lines. Additionally, the fluorescent properties of the sensor, allow for applications in high-throughput screening approaches, which could be of interest in the context of drug development.

- The authors do not mention that DPP8/9 inhibition with 1G244 can induce cell death via the inflammasome, which might influence the interpretation of many results in this manuscript. Please discuss the available literature on the effects of DPP8/9 inhibition on cell viability in the different experiments of the manuscript.

We assessed cell viability over time upon treatment with different amounts of 1G244. Viability of HEK293 and melanoma cell models remained comparable to untreated cells for the treatment regimens used in this study (**new Figures S1B, S2A, S6C**). Conversely, B cells require DPP8/9 activity for maturation and are sensitive to DPP8/9 loss also when unstimulated (**new Figures S8, S9**). The viability and cell death modalities found in stimulated and unstimulated B-cells are now extensively characterized in the manuscript (**Figure 6, Figure S7, new Figures S8-S9; see also answer to comment from referee #3 below**). Specifically, we found that at concentrations starting from 4 μ M 1G244 numbers of live B-cells dropped in the course of LPS treatment due to apoptosis, indicated by Annexin V binding and sub-G1 cell cycle arrest, but not pyroptosis.

These results are now presented in the manuscript.

Reviewer #3 (Comments to the Authors (Required)):

Summary:

The authors developed a genetically encoded fluorescent DPP8/9 activity sensor that can be used for in-cell activity measurement of these two proteases. For the development of this sensor, they exploited a previously identified DPP9 substrate, AK2. Briefly, N-terminal processing of AK2 by methionine aminopeptidases reveals the DPP9 cleavage site (an Ala-Pro N-terminus). Removal of the AP dipeptide exposes an unstable N-terminal residue, which in turn leads to AK2 proteasomal degradation by the N-end rule pathway. By fusing the first 15 residues of AK2 to GFP (AK2 1-15 GFP), they can monitor DPP8/9 activity by measuring GFP fluorescence (ie inhibition of DPP8/9 gives greater fluorescence). As a control, they performed a single point mutation on the AK2 N-terminus, S4P, (AK2 1-15 S4P GFP), which prevents DPP8/9 cleavage. By combining these two ORFs into one plasmid, separated by an IRES site they can monitor DPP8/9 activity by measuring the stability of the GFP reporter without being confounded by expression levels of the reporter. The authors provide appropriate controls for this reporter and verify that is DPP8/9 specific. They show that their reporter can be stabilized by 1G244, a DPP8/9 inhibitor which prevents AP processing and by MG132, a proteasome inhibitor, which prevents GFP degradation following the exposure of the N-degron. Importantly, the fluorescence of this reporter can be assessed by three independent methods: by fluorescence microscopy, flow cytometry and fluorescence plate reader, allowing its application in multiple cell lines and experiments set ups. Overall, this is an well-designed and useful DPP8/9 activity in-cell reporter. However, the application of the reporter in Figure 6 has major issues, as all of the assays are could simply be observing DPP8/9 induced pyroptosis (ie cell death counfounding their interpretation) .

We thank this reviewer for the comments. As this reviewer noted correctly, 1G244 appears to elicit cell death in activated B cells, which we already noted (please see Supplementary Figure 5a of the original submission which is now the **new Figure S7A**). Because we did observe an increase in the subG1 DNA peak in response to 1G244 treatment, we suspected that DPP8/9 inhibition elicits apoptosis in the activated B cells. This reviewer argues for Caspase-1 mediated pyroptosis.

In summary, the experiments that we conducted do not point to a role of 1G244 in pyroptosis. We conclude that DPP8/9 are more active in activated B cells than in plasmablasts. Thus, DPP8/9 inhibition merely affects activated B cells rather than established plasmablasts/plasma cells (see original submission, **Figure 6**). This finding was again reproduced by assessing absolute cell numbers in 4 independent cultures (see **new Figure S9**). Hence, we propose that the function of DPP8/9 is to maintain mitochondrial function and integrity in activated B cells, thereby, supporting cell growth and survival independent of pyroptosis. This notion aligns with increased DPP8/9 activity in activated B cells, as measured by expression of our new DiPaK sensor in those cells.

Major comments:

Figure 6: DPP8/9 inhibition can induce pyroptosis in resting lymphocytes, including T cells and B cells (Johnson, et al. 2020, Linder et al. 2020). It seems likely that 1G244 is inducing cell death in Fig 6B-D, thereby explaining their phenotypic responses. The authors should:

1. Include cell viability assays (Cell-titer GLO or LDH release assays) for the cells treated with the different concentrations of 1G244. This is not only important for controlling for DPP8/9 inhibitor-induced pyroptosis, but also for controlling for 1G244 off-target-mediated cell death (Okondo et al, 2017).

As suggested, we included viability assays by performing Annexin V staining along with DNA staining to assess Phosphatidylserine exposure, and we repeated the measurements of the DNA content in 1G244 treated cells. We chose a concentration of 8 μM (see previous Supplementary Figure 5a which is now the **new Figure S7A**). In line with our previous experiments, 1G244 treatment below 10 μM induced apoptosis as judged by enhanced PS exposure and DNA degradation (**new Figure S9**).

2. Perform the same experiments in the presence of a caspase-1 inhibitor (VX-765).

We repeated the experiments with appropriate concentrations of the Caspase 1 inhibitor VX-765. VX-765 did not exert any effect on the activated B cells, either alone, or in combination with 1G244. These data suggest that 1G244 does not induce pyroptosis in LPS activated B cells, and that pyroptosis does not play a major role in survival or differentiation of LPS activated B cells (**new Figures 8 and 9**).

3. Figure 6B and S5A: It would be nice to include a DiPAK ratio measurement. Does the decrease in B cell growth correlate with DPP8/9 activity as measured by the DiPAK ratio in the presence of a caspase-1 inhibitor?

The DiPAK ratio in response to the 1G244 treatment is depicted in **Figure 6E**. Because VX-765 has no effects that are comparable to 1G244 whatsoever, we did not measure the DiPAK ratio in VX-765 treated cells.

4. Figure 6E: it seems like there really isn't a DPP8/9 activity difference in activated B cells and plasmablasts. Moreover, it seems like there shouldn't really be any DPP8/9 activity at 8 μM 1G244. Is this figure scaled to make a very minor difference seem large?

The statistic clearly tells that there is more DiPAK activity in activated B cells than in plasmablasts. The DiPAK ratio increases ~ 7 fold in activated B cells compared to ~ 4 fold in plasmablasts reflecting differences in DPP8/9 activity. We believe that the figures are scaled appropriately (compare insets).

5. 1G244 has DPP8/9 inhibition-independent effects at concentrations >10 μ M (Okondo et al, 2017), which the authors use in Figure 6. The authors should use one completely unrelated DPP8/9 inhibitor (VbP) to show that the effect is really due to DPP8/9 inhibition.

We did the new analyses with a 1G244 concentration of 8 μ M (IC50 for DPP8: 12nM, DPP9: 84 nM). We are thus confident that the observed effects are not off-target as the off-target effects are reported at >10 μ M. We used PT-100 (ValBoroPro) accordingly at a concentration of 2 μ M, adjusted to its IC50 for DPP8/9 (4 nM and 11 nM, respectively). It has been reported that PT-100 does induce Pyroptosis; however, we did not see any effect of PT-100. This is in line with the absence of an effect of VX-765. While PT-100 may inhibit DPP8/9, it might only do so in B cells at much higher concentrations. (new Figure S8,9). Following these findings we removed data involving treatments with 1G244 concentrations > 8 μ M from the manuscript to prevent confusion.

Minor comments:

Figure 3A: would be nice to include the brightfield versions of the respective fluorescence microscopy image

We repeated the experiment in Figure 3A including controls without DiPAK expression and replaced the former panel by this dataset (Figure 3A). Additionally, we provide brightfield images, see below:

April 18, 2025

RE: Life Science Alliance Manuscript #LSA-2024-03076-TR

Prof. Jan Riemer
University of Cologne
Biochemistry
Zuelpicher Str. 47a
Cologne 50674
Germany

Dear Dr. Riemer,

Thank you for submitting your revised manuscript entitled "A fluorescent sensor for real-time monitoring of DPP8/9 reveals crucial roles in immunity and cancer". As you will see, Reviewers 1 and 2 remarked that most major concerns have been resolved in this revision. Reviewer 1 remarked that the influence of DPP4 on the response of DiPAK was not explicitly shown. While experimental data to resolve this is not required, please include a statement to acknowledge this limitation. Reviewer 3 remained concerned on the potential of pyroptosis in B cells in this study. However in comments to the editors Reviewers 1 and 2 both noted this does not need to be resolved for publication, and we concur with their view that these questions are beyond the scope of the current work. We would be happy to publish your paper in Life Science Alliance pending the above text change and final revisions necessary to meet our formatting guidelines.

- please upload all figure files as individual ones, including the supplementary figure files; all figure legends should only appear in the main manuscript file.
- please add ORCID ID for corresponding author - you should have received instructions on how to do so.
- please add the X and Bluesky handles of your host institute/organization as well as your own or/and one of the authors in our system.
- please upload your Tables in editable .doc or excel format;
- please add your main, supplementary figure, and table legends to the main manuscript text after the references section.
- please incorporate supplemental references into the main references list.
- please be sure that the authorship listing and order is correct and that matches between the system and manuscript file.
- please use the [10 author names, et al.] format in your references (i.e., limit the author names to the first 10).
- please add callouts for Figures S6A, C; S7A; S8A-F; S9A-B to your main manuscript text.
- please ensure the Data Availability statement is complete.

A. FINAL FILES:

- An editable version of the final text (.DOC or .DOCX) is needed for copyediting (no PDFs).
- High-resolution figure, supplementary figure and video files uploaded as individual files: See our detailed guidelines for preparing your production-ready images, <https://www.life-science-alliance.org/authors>
- Summary blurb (enter in submission system): A short text summarizing in a single sentence the study (max. 200 characters)

including spaces). This text is used in conjunction with the titles of papers, hence should be informative and complementary to the title. It should describe the context and significance of the findings for a general readership; it should be written in the present tense and refer to the work in the third person. Author names should not be mentioned.

B. MANUSCRIPT ORGANIZATION AND FORMATTING:

Sincerely,

Reviewer #1 (Comments to the Authors (Required)):

Many issues have been addressed well.

The first query I provided on this ms has not been addressed. This ms needs to show whether DiPAK also detects/measures DPP4 or PREP activity to some extent. This is important and relevant when cell types other than HEK293 are used by researchers who read this ms and want to use DiPAK. HEK293, in my experience, contains very little DPP4, but many cell lines and cells are high in PREP and/or DPP4.

The key experiment that must be done is to show whether DiPAK responds to DPP4. This must be done on cells in which DPP4 is readily measurable. Transfecting in DPP4 is best for performing a dose-response of DiPAK on a range of quantities of DPP4.

The comment I made 'In addition', of using a DPP4-selective inhibitor such a sitagliptin and a PREP-selective inhibitor, was done. But the above experiments on DPP4+ cells, which were not done, are the more important ones. Do HEK293 contain PREP? If so, the inhibitor expt is sufficient, and it appears to be sufficient for PREP.

Reviewer #2 (Comments to the Authors (Required)):

The authors have thoroughly revised and extended their study on DiPak as a tool to further study DPP8/9 activity. Most of my questions are answered adequately.

Reviewer #3 (Comments to the Authors (Required)):

The authors have not addressed the major concern expressed in the previous round of reviews - that the effects observed in B cells are due to 1G244-induced cell death. Moreover, this death appears to not even involve the inhibition of DPP8/9 (which causes inflammasome activation and pyroptosis), but is due to the previously established off-target toxicity of 1g244.

The compound that the authors rely on, 1G244, is not a selective DPP8/9 inhibitor. At concentrations about 10 μ M (depending on the cell type), it induces a DPP8/9-, caspase-1-independent form of cell death (Okondo et al, Nat Chem Bio, 2017, Fig. 3C). Other DPP8/9 inhibitors like Val-boroPro (PT-100) do not activate this form of cell death (see the same figure). The authors here find in Fig. S8 that 1G244 at 8 μ M indeed induces cell death that is likely not DPP8/9 dependent, as PT-100 does not cause a similar cell death. The conclusions drawn using 1G244 at 8 μ M in these cells is therefore likely not due to DPP8/9 inhibition.

The major point of this paper is to introduce a probe to monitor DPP8/9 activity in cells. They should use this probe to show that the unrelated DPP8/9 inhibitor (e.g., PT-100) similarly inhibits DPP8/9 in cells (it certainly will at 2 μ M, based on work from many labs). This result will formally show that their B cell death is all unrelated to DPP8/9 inhibition and is due to the established off target of 1g244. This entire figure and supporting data should then be removed from the paper, and they key experiments that show their probe works in cells should be repeated using PT-100 in addition to 1G244.

April 30, 2025

RE: Life Science Alliance Manuscript #LSA-2024-03076-TRR

Prof. Jan Riemer
University of Cologne
Biochemistry
Zuelpicher Str. 47a
Cologne 50674
Germany

Dear Dr. Riemer,

Thank you for submitting your Methods entitled "A fluorescent sensor for real-time monitoring of DPP8/9 reveals crucial roles in immunity and cancer". It is a pleasure to let you know that your manuscript is now accepted for publication in Life Science Alliance. Congratulations on this interesting work.

DISTRIBUTION OF MATERIALS:

Again, congratulations on a very nice paper. I hope you found the review process to be constructive and are pleased with how the manuscript was handled editorially. We look forward to future exciting submissions from your lab.

Sincerely,
